



**Population-specific responses in physiological rates of *Emiliania huxleyi* to a**
**broad CO$_2$ range**
**Yong Zhang,[1,5,*] Lennart T. Bach,[1] Kai T. Lohbeck,[1,2,6] Kai G. Schulz,[3] Luisa**
**Listmann,[2] Regina Klapper,[4] Ulf Riebesell[1]**
[1]Biological Oceanography, GEOMAR Helmholtz-Centre for Ocean Research Kiel,
Kiel, Germany
[2]Evolutionary Ecology of Marine Fishes, GEOMAR Helmholtz-Centre for Ocean
Research Kiel, Kiel, Germany
[3]Centre for Coastal Biogeochemistry, School of Science, Environment and
Engineering, Southern Cross University, Lismore, NSW, Australia
[4]Goethe-University, Institute for Ecology, Evolution and Diversity; Senckenberg
Gesellschaft für Naturforschung, Senckenberg Biodiversity and Climate Research
Centre, Frankfurt am Main, Germany
[5]State Key Laboratory of Marine Environmental Science, College of Ocean and Earth
Sciences, Xiamen University (Xiang-An Campus), Xiamen 361102, China
[6]Department of Marine Sciences, University of Gothenburg, Gothenburg, Sweden
Running head: *population response of* Emiliania huxleyi *to CO$_2$*
[*]Correspondence to: Yong Zhang (zhangyong1983@xmu.edu.cn)
Keywords: CO$_2$; coccolithophore; physiological rate; population; strain



**Abstract**
Although coccolithophore physiological responses to $CO_2$-induced changes in
seawater carbonate chemistry have been widely studied in the past, there is limited
knowledge on the variability of physiological responses between populations. In the
present study, we investigated the population-specific responses of growth, particulate
organic (POC) and inorganic carbon (PIC) production rates of 17 strains of the
coccolithophore *Emiliania huxleyi* from three regions in the North Atlantic Ocean
(Azores, Canary Islands, and Norwegian coast near Bergen) to a $CO_2$ partial pressure
($p$CO$_2$) range from 120 µatm to 2630 µatm. Physiological rates of each population
and individual strain displayed the expected optimum curve responses to the $p$CO$_2$
gradient. Optimal $p$CO$_2$ for growth and POC production rates and tolerance to low pH
(i.e. high proton concentration) was significantly higher in an *E. huxleyi* population
isolated from a Norwegian fjord than in those isolated near the Azores and Canary
Islands. This may be due to the large $p$CO$_2$ and pH variability in coastal waters off
Bergen compared to the rather stable oceanic conditions at the other two sites.
Maximum growth and POC production rates of the Azores and Bergen populations
were similar and significantly higher than of the Canary Islands population. One of
the reasons may be that the chosen incubation temperature (16 $^o$C) is slightly below
what strains isolated near the Canary Islands normally experience. Our results indicate
adaptation of *E. huxleyi* to their local environmental conditions. Within each
population, different growth, POC and PIC production rates at different $p$CO$_2$ levels
indicated strain-specific phenotypic plasticity. The existence of distinct carbonate





chemistry responses between and within populations will likely benefit *E. huxleyi* to
acclimate to rising $CO_2$ levels in the oceans.






















## 1 Introduction

Coccolithophores form a layer of calcium carbonate ($CaCO_3$) platelets (coccoliths) around their cells. Coccoliths are of biogeochemical importance due to ballasting of organic matter with $CaCO_3$, a phenomenon which is thought to promote the transport of organic carbon to the deep ocean (Klaas and Archer, 2002; Rost and Riebesell, 2004). The coccolithophore *Emiliania huxleyi* forms extensive blooms under favourable light intensity, temperature and nutrient conditions, with different morphotypes in certain regions (Cook et al., 2011; Henderiks et al., 2012; Smith et al., 2012; Balch et al., 2014).

Variable responses of growth, photosynthetic carbon fixation and calcification rates of different *E. huxleyi* strains to rising $CO_2$ levels have been reported (Langer et al., 2009; Hoppe et al., 2011; Müller et al., 2015; Hattich et al, 2017) and are likely a result of intra-specific variability of genotypes (Langer et al., 2009). Several recent studies observed optimum curve responses in physiological rates of a single *E. huxleyi* strain to a broad $pCO_2$ range from about 20 µatm to 5000 µatm, and linked them to inorganic carbon substrate limitation at low $pCO_2$ and inhibiting $H^+$ concentrations at high $pCO_2$ (Bach et al., 2011; 2015; Kottmeier et al., 2016). Until now, studies on the physiological responses of *E. huxleyi* to rising $CO_2$ are mostly based on a few genotypes and little is known about the potential variability in $CO_2$ and $H^+$ sensitivity between and within populations. Recently, several studies found substantial variations in $CO_2$ responses for $N_2$ fixation rates between *Trichodesmium* strains, as well as for





growth rates between strains of *Gephyrocapsa oceanica*, *Ostreococcus tauri* and
*Fragilariopsis cylindrus* (Hutchins et al., 2013; Schaum et al., 2013; Pancic et al.,
2015; Hattich et al., 2017). These indicate that multiple strains should be considered
for investigating phytoplankton responses to climate change (Zhang et al., 2014;
Blanco-Ameijeiras et al., 2016).
Oceanographic boundaries formed by both ocean currents and environmental
factors such as temperature, can limit dispersal of marine phytoplankton, reduce gene
flow between geographic populations, and give rise to differentiated populations
(Palumbi, 1994). Different populations were found to show different growth rates for
*E. huxleyi*, *G. oceanica*, and *Skeletonema marinoi* at the same temperatures, and for
*Ditylum brightwellii* at the same light intensities (Brand, 1982; Rynearson and
Armbrust, 2004; Kremp et al., 2012; Zhang et al., 2014). Phenotypic plasticity
describes the ability of a strain to change its morphology or physiology in response to
changing environmental conditions (Bradshaw, 1965). Plasticity can be assessed by
analyzing the reaction norm of one trait and a plastic response may allow a strain to
acclimate to environmental change (Reusch, 2014; Levis and Pfennig, 2016).
In order to better understand how local adaptation affects the physiological
response of *E. huxleyi* to rising $CO_2$ conditions, we isolated 17 strains from three
regions in the Atlantic Ocean, and assessed growth, carbon fixation and calcification
responses of the population over a $p$CO$_2$ range from 120 µatm to 2630 µatm.

**2 Materials and methods**



### 2.1 Cell isolation sites and experimental setup

*Emiliania huxleyi* strains EHGKL B95, B63, B62, B51, B41 and B17 originated from
Raunefjord (Norway 60$^{o}$18'N, 05$^{o}$15'E) and were isolated by K. T. Lohbeck in May,
2009 (Lohbeck et al., 2012) at ~ 10 $^{o}$C in-situ water temperature. *E. huxleyi* strains
EHGLE A23, A22, A21, A19, A13 and A10 originated from coastal waters near the
Azores (38$^{o}$34'N, 28$^{o}$42'W) and were isolated by S. L. Eggers in May or June, 2010
at ~ 17 $^{o}$C in-situ water temperature. *E. huxleyi* strains EHGKL C98, C91, C90, C41
and C35 originated from coastal waters near Gran Canaria (27$^{o}$58'N, 15$^{o}$36'W) and
were isolated by K. T. Lohbeck in February, 2014 at ~ 18 $^{o}$C in-situ water temperature.
Seasonal $CO_2$ concentration in the surface seawater ranges from 240 µatm to 400
µatm near Bergen, from 320 µatm to 400 µatm around the Azores and from 320 µatm
to 400 µatm around the Canary Islands (Table 1). Monthly surface seawater
temperature ranges from 6.0 to 16.0 $^{o}$C near Bergen, 15.6 to 22.3 $^{o}$C around the
Azores and from 18.0 to 23.5 $^{o}$C around the Canary Islands (Table S1).

All 17 strains belong to morphotype A and have been deposited at the Roscoff
culture collection (RCC) under the official names as shown above. Genetically
different isolates, here called strains, were identified by 5 microsatellite markers
(P02E09, P02B12, P02F11, EHMS37, EHMS15) (Table S2). For a description of
primer testing, deoxyribonucleic acid (DNA) extraction, DNA concentration
measurements, and polymerase chain reaction (PCR) protocols see Zhang et al.
(2014). The Azores and Bergen strains had been used earlier by Zhang et al. (2014).

The six or five (in case of Canary Islands) strains of each region were used to test
the physiological response to varying $CO_2$ concentrations at constant total alkalinity
(TA). The experiment was performed in six consecutive incubations, with one strain
from each population (Azores, Bergen, Canary Islands) being cultured at a time.
Monoclonal populations were always grown in sterile-filtered (0.2 μm diameter,
Sartobran® P 300, Sartorius) artificial seawater medium (ASW) as dilute batch
cultures at 200 μmol photons $m^{-2}$ $s^{-1}$ light intensity under a 16/8 h light/dark cycle
(light period: 5:00 a.m to 9:00 p.m.) at 16 $^o$C which we consider to be the best
compromise for the three different origins of the strains. Nutrients were added in
excess (with nitrate and phosphate concentrations of 64 μmol $kg^{-1}$ and 4 μmol $kg^{-1}$,
respectively). For the preparation of ASW and nutrient additions see Zhang et al.
(2014). Calculated volumes of $Na_2CO_3$ and hydrochloric acid were added to the ASW
to achieve target $CO_2$ levels at an average total alkalinity (TA) of 2319 $\pm$ 23 μmol $kg^-$
$^1$ (Pierrot et al., 2006; Bach et al., 2011). Each strain was grown under 11 $CO_2$ levels
ranging from 115 μatm to 3070 μatm without replicate. Mean response variables of all
strains with a population were calculated and mean $CO_2$ levels of all strains within a
population ranged from 120 μatm to 2630 μatm. Cells grew in the experimental
conditions for at least 7 generations, which corresponded to 4–7 days depending on
cell division rates. Cells were cultured for 4 days in 120–925 μatm $CO_2$, for 5 days in
1080–1380 μatm $CO_2$, and for 6 or 7 days in 1550–2630 μatm $CO_2$. Initial cell
concentration was 200 cells $ml^{-1}$ and final cell concentration was lower than 100,000
cells $ml^{-1}$. Dissolved inorganic carbon (DIC) concentrations and $p$CO$_2$ levels changed





less than 7% and 11%, respectively, during the experimental growth phase.

**2.2 $pH_T$ and total alkalinity measurements**
At 10:00 a.m. on the last day of incubations (at day 4–7 depending on $CO_2$
concentration), $pH_T$ and TA samples were filtered (0.2 μm diameter, Filtropur S 0.2,
Sarstedt) by gentle pressure and stored at $4^oC$ for a maximum of 14 days. The entire
sampling lasted less than 2 h. The $pH_T$ sample bottles were filled with considerable
overflow and closed tightly with no space. $pH_T$ was measured spectrophotometrically
(Cary 100, Agilent) using the indicator dye *m*-cresol purple (Sigma-Aldrich) similar
to Carter et al. (2013) with constants of acid dissociation for the protonated and un-
protonated forms reported in Clayton and Byrne (1993). TA was measured by open-
cell potentiometric titration (862 Compact Titrosampler, Metrohm) according to
Dickson et al. (2003). The carbonate system was calculated from measured TA, $pH_T$,
(assuming 4 μmol $kg^{-1}$ of phosphate and 0 μmol $kg^{-1}$ of silicate) using the CO2
System Calculations in MS Excel software (Pierrot et al., 2006) with carbonic acid
constants $K_1$ and $K_2$ as determined by Roy et al. (1993).

**2.3 Growth rate measurements**
At 1:00 p.m. on the last day of incubation, 25 ml samples were used to measure cell
concentration. Cell concentration was determined within two hours using a Z2 Coulter
Particle Counter (Beckman). Growth rate (μ) was calculated according to:
$$\mu = (\ln N_1 - \ln N_0) / d \qquad (1)$$



where $N_1$ is cell concentration on the last day of incubation, $N_0$ is 200 cells mL$^{-1}$, and
$d$ is the time period for growth of algae in days.

**2.4 Particulate organic (POC) and inorganic (PIC) carbon measurements**
At 3:00 p.m. on the last day of incubation, cells for total particulate (TPC) and total
organic (TOC) carbon were filtered onto GF/F filters which were pre-combusted at
500 $^{o}$C for 8 h. Samples of background particulate carbon (BPC) were determined in a
similar way but using filtered ASW without algae, which was previously adjusted to
target $p$CO$_2$ levels, and allowed to age for about 7 days under incubation conditions
(*see* above). All samples were placed at –20$^{o}$C. BPC filters were used as blanks to
correct for organic carbon in the medium. TOC and BPC filters were acid fumed.
Afterwards, all filters were dried for 8 h at 60$^{o}$C. TPC, TOC and BPC were measured
using an Elemental Analyzer (EuroEA, Hekatech GmbH). The percentages of BPC in
TPC were about 20% at cell densities < 10,000 cells ml$^{-1}$ and about 10% at cell
densities > 40,000 cells ml$^{-1}$. POC was calculated as the difference between TOC and
BPC. PIC was calculated as the difference between TPC and TOC. POC and PIC
production rates were calculated as:
$$\text{POC production rate} = \mu \ (\text{d}^{-1}) \times (\text{TOC} - \text{BPC}) \ (\text{pg C cell}^{-1}) \quad (2)$$
$$\text{PIC production rate} = \mu \ (\text{d}^{-1}) \times (\text{TPC} - \text{TOC}) \ (\text{pg C cell}^{-1}) \quad (3)$$

**2.5 Data analysis**
The nonlinear regression model (4) was used to fit growth, POC and PIC production



rates yielding theoretical optimum $p\text{CO}_2$ and maximum values for each of the three
populations (combining the data of five or six strains) (Bach et al., 2011).
$$y = \frac{X \times pCO_2}{Y + pCO_2} - s \times pCO_2 \qquad (4)$$
where $X$ and $Y$ are fitted parameters, and $s$ is the sensitivity constant which indicates
the effect of rising $\text{H}^+$. Based on the fitted $X$, $Y$ and $s$, we calculated the $p\text{CO}_2$ optima
($K_m$) for physiological rates according to equation (5). Maximum growth, POC and
PIC production rates were calculated by using equation (4) based on $K_m$.
$$K_m = \sqrt{\frac{X \times Y}{s}} - Y \qquad (5)$$
The relative values for growth, POC and PIC production rates were calculated as
ratios of growth, POC and PIC production rates at each $p\text{CO}_2$ level to the maximum
(highest) rates. We obtained the relative sensitivity constant by fitting function (4)
based on relative growth, POC and PIC production rates.

211        A one-way ANOVA was then used to test for statistically significant differences in

theoretical optimum $p\text{CO}_2$, maximum value and relative sensitivity constant between
populations. A Tukey HSD test was conducted to determine the differences between
strains from different populations. A Shapiro–Wilk's analysis was tested to analyze
residual normality. Statistical calculations were carried out using $R$ and significance
was shown by $p < 0.05$.

**3  Results**

**3.1  Carbonate chemistry parameters**

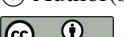



Carbonate system parameters are shown in Table 2. Average $p$CO$_2$ levels of the ASW
ranged from 125 µatm to 2490 µatm for the Azores population, from 120 µatm to
2280 µatm for the Bergen population, and from 130 µatm to 2630 µatm for the
Canary Islands population. Corresponding pH$_T$ values of the ASW ranged from 8.46
to 7.33 for the Azores population, from 8.47 to 7.37 for the Bergen population, and
from 8.45 to 7.31 for the Canary Islands population.

**3.2 Measured growth, POC and PIC production rates of each population**
Growth rates, POC and PIC production rates of the three *E. huxleyi* populations
increased with rising $p$CO$_2$, reached a maximum, and then declined with further $p$CO$_2$
increase (Fig. 1). Growth rates of the Azores and Bergen populations were larger than
those of the Canary Islands population at all investigated $p$CO$_2$ levels (Fig. 1a). With
rising $p$CO$_2$ levels beyond the $p$CO$_2$ optimum, decline in growth rates was more
pronounced in the Azores and Canary Islands populations than in the Bergen
population (Fig. 1b).

236       Measured POC production rates of the Azores and Bergen populations were larger

than those of the Canary Islands population at all $p$CO$_2$ levels (Fig. 1c) and decline in
POC production rates with increasing $p$CO$_2$ levels beyond the $p$CO$_2$ optimum was
larger in the Azores and Canary Islands populations than in the Bergen population
(Fig. 1d).

241       Measured PIC production rates at investigated $p$CO$_2$ levels did not show significant

differences among the Azores, Bergen and Canary Islands populations (Fig. 1e).



Exceptions were that at 365–695 µatm, PIC production rates of the Azores population
were larger than those of the Canary Islands population (all $p < 0.05$).

**3.3  Physiological responses of populations to $p\text{CO}_2$**
Calculated optimum $p\text{CO}_2$ for growth, POC and PIC production rates of the Bergen
population were significantly larger than those of the Azores and Canary Islands
populations (all $p < 0.05$) (Fig. 2a–c). Optimum $p\text{CO}_2$ for these physiological rates
between the Azores and Canary Islands population were not different (all $p > 0.1$).

251        Calculated maximum growth rates, POC and PIC production rates were not

significantly different between the Azores and the Bergen populations (all $p > 0.1$)
(Fig. 2d–f). Maximum growth rate and POC production rate of the Canary Islands
population were significantly lower than those of the Azores and Bergen populations
(both $p < 0.01$) (Fig. 2d,e). Maximum PIC production rates of the Canary Islands
population were significantly lower than that of the Azores population ($p < 0.05$),
while there was no difference to the Bergen population ($p > 0.1$) (Fig. 2f).

258        Fitted relative sensitivity constants for growth and POC production rates of the

Bergen population were significantly lower than those of the Azores and Canary
Islands populations ($p < 0.01$) (Fig. 2g, h). Fitted relative sensitivity constants for
growth and POC production rates between the Azores and Canary Islands populations
were not significantly different ($p > 0.1$). Fitted relative sensitivity constants for PIC
production rates did not show difference among three populations ($p = 0.13$) (Fig. 2i).




**3.4 Physiological responses of individual strains to $p$CO$_2$**
Measured growth rates, POC and PIC production rates of 17 *E. huxleyi* strains showed
optimum curve response patterns to the broad $p$CO$_2$ gradient (Fig. 3). Variations in
calculated $p$CO$_2$ optima, maximum values and relative sensitivity constants of
physiological rates were found between the strains (Table 3).
For all strains within each population, optimum $p$CO$_2$ of POC production rates
were larger than optimum $p$CO$_2$ of growth rates or PIC production rates with the
exception of optimum $p$CO$_2$ of POC and PIC production rates of *E. huxleyi* strain
EHGLE A22 (Table 3). Compared to the Azores and Bergen populations, strains
isolated near the Canary Islands showed larger variation in optimum $p$CO$_2$ of PIC
production rates. Within the Azores population, variations in maximum values ($V_{max}$)
and relative sensitivity constants ($rs$) of growth, POC and PIC production rates of all
strains were larger than those within the Bergen and Canary Islands populations (Fig.

278 3).


**4 Discussion**

We investigated growth, POC and PIC production rates of 17 *E. huxleyi* strains from
three populations to a broad $p$CO$_2$ range (120–2630 µatm). The three populations
differed significantly in growth and POC production rates at the investigated $p$CO$_2$
levels. The reaction norms of the individual strains and populations equaled an
optimum curve for all physiological rates (Figs. 1 and 3). However, we detected




distinct $pCO_2$ optima for growth, POC and PIC production rates, and different $H^+$
sensitivities for growth and POC production rates among them (Fig. 2). These results
indicate the existence of distinct populations in the cosmopolitan coccolithophore *E.*
*huxleyi*.

291        In comparison to the Azores and Canary Islands populations, variability in growth

rates between strains of the Bergen population was smaller even though they had
higher growth rates at all $pCO_2$ levels (Fig. 3). Furthermore, the Bergen population
showed significantly higher $pCO_2$ optima and lower $H^+$ sensitivity for growth and
POC production rates (Fig. 2). These findings indicate that the Bergen population may
be more tolerant to changing carbonate chemistry in terms of its growth and
photosynthetic carbon fixation rates. The Bergen strains were isolated from coastal
waters, while the Azores and Canary Islands strains were isolated from a more
oceanic environment. Seawater carbonate chemistry of coastal waters is usually more
dynamic than in the open ocean (Cai, 2011). In fact, previous studies have reported
that $CO_2$ and pH variability of the seawater off Bergen was larger than off the Azores
and Canary Islands (Table 1). Doblin and van Sebille (2016) suggested that
phytoplankton populations should be constantly under selection when experienced
with changing environmental conditions. In this case, the Bergen population, exposed
to larger $CO_2$ or pH fluctuations, may have acquired a higher capacity to acclimate to
changing carbonate chemistry resulting in a higher tolerance (or lower sensitivity) to
rising $CO_2$ levels. In contrast, the Azores and Canary Islands populations experience
similar, less variable seawater carbonate chemistry conditions in their natural





environment, which could explain why they also show similar $p\mathrm{CO}_2$ optima and $\mathrm{H}^+$
sensitivity for physiological rates (Fig. 2).
In an earlier study (Zhang et al., 2014), growth rates of the same Azores and
Bergen strains as used here were measured at 8–28 $^{\mathrm{o}}$C. While at 26–28 ℃ the Bergen
strains grew slower than the Azores strains, at 8 ℃ the Azores strains grew slower
than the Bergen strains. This illustrates nicely that local temperature adaptation can
significantly affect growth of *E. huxleyi* strains in laboratory experiments.
Considering these findings and the temperature ranges of three isolated locations
(Table S1), the incubation temperature of 16 ℃ used in the present study was lower
than the minimum sea surface temperature (SST) commonly recorded at the Canary
Islands. In contrast, SSTs of 16 ℃ and lower have been reported for Azores and
Bergen waters (Table S1). When exposed to 16 $^{\mathrm{o}}$C, growth rate of the Canary Islands
population might have been already below their optimum and thus it grew slower than
the other populations (Fig. 2d). One of the reasons may be that compared to the
Azores and Bergen populations, 16 $^{\mathrm{o}}$C likely causes lower the carbon uptake and
carbon-use efficiency of the Canary Islands population (Sett et al., 2014). Thus, with
rising $\mathrm{CO}_2$, growth, photosynthetic carbon fixation and calcification rates of the
Canary Islands population cannot increase as much as in the Azores and Bergen
populations.
Before we started this experiment, strains isolated from the Azores, Bergen and
Canary Islands grew as stock cultures at 15 $^{\mathrm{o}}$C and 400 μatm for 4 years, 5 years and
3 months, respectively. Schaum et al. (2015) provide evidence that long-term



laboratory incubation affects responses of phytoplankton to different $p\mathrm{CO_2}$ levels.
Thus, it is conceivable that the same selection history in the laboratory incubation
may contribute to a more similar response of growth, POC and PIC production rates
between the Azores and Bergen populations at low $p\mathrm{CO_2}$ levels (Fig. 1).
Our results indicate that *E. hulxyei* populations are adapted to the specific
environmental conditions of their origin, resulting in different responses to increasing
$p\mathrm{CO_2}$ levels. The ability to adapt to diverse environmental conditions is reflected in
the global distribution of *E. huxleyi* (Paasche, 2002), spanning a temperature range of
about 30 $^\mathrm{o}$C. In natural seawater, due to ocean currents and gene flow, populations at
any given location may get replaced by populations transported there from other
locations when having a higher potential to adapt to a changing environment (Doblin
and van Sebille, 2016). In addition, *E. huxleyi* take up $\mathrm{HCO_3^-}$ to calcify and generate
proton, and increase in proton concentration may mitigate the potential of the ocean to
absorb atmospheric $\mathrm{CO_2}$ (Paasche, 2002). Thus, due to population-specific growth
and PIC production rates or quotas, changes in species composition, corresponding
changes in PIC productions, may affect the ability of the ocean to take up $\mathrm{CO_2}$.
Within a population, individual strains showed different growth, POC and PIC
production rates at different $p\mathrm{CO_2}$ levels, indicating phenotypic plasticity of
individual strains (Reusch, 2014). Phenotypic plasticity constitutes an advantage for
individual strains to adapt to elevated $p\mathrm{CO_2}$ by changing their fitness-relevant traits
(Schaum et al., 2013). Additionally, our results also suggest that strain-specific PIC
quota may be the basis of variation in coccoliths of *E. huxleyi* within the morphotype



A (Fig. S3) (Young, 1994; Paasche, 2002).
The strain-specific $CO_2$-response curves revealed considerable physiological
diversity in co-occurring strains (Fig. 3). Physiological variability makes a population
more resilient and increases its ability to persist in variable environments (Gsell et al.,
2012; Hattich et al., 2017). It is clear that other environmental factors such as light
intensity, temperature and nutrient concentration affect the responses of physiological
rates of individual *E. huxleyi* strains to changing carbonate chemistry, and thus change
the physiological variability within populations (Zhang et al., 2015; Feng et al., 2017).
However, different sensitivities and requirements of each strain to the variable
environments can allow strains to co-exist within a population in the natural
environment (Hutchinson, 1961; Reed et al., 2010; Krueger-Hadfield et al., 2014). In
changing oceans, strain succession is likely to occur and shift the population
composition (Blanco-Ameijeiras et al., 2016; Hattich et al., 2017). Strains with high
growth rates may outcompete other strains in the oceans (Schaum et al., 2013).
Significant positive correlation between growth and POC production rate or POC
quota (Fig. 4S) suggests that the dominated strains can also take up dissolved
inorganic carbon faster from the oceans or fix carbon faster. This may increase the
potential of the oceans to absorb $CO_2$ from the atmosphere or the carbon storage
capacity of the oceans when large *E. huxleyi* blooms occur (Blanco-Ameijeiras et al.,
2016), which will mitigate rising $CO_2$ levels in the atmosphere.

**5 Conclusions**



In the present study, we found population-specific responses in physiological rates of
*E. huxleyi* to a broad $pCO_2$ range, which may have arisen from local adaptation to
environmental conditions at their origins. Our results suggest that when assessing
phytoplankton responses to changing environments on a global scale, variability in
population or strain responses need to be considered.





















*Author contributions*. YZ, LTB, UR designed the experiment. YZ, LL, RK performed
the experiment. YZ prepare the manuscript and all authors analysed the data,
reviewed and improved the manuscript.


*Competing interests*. The authors declare that they have no conflict of interest.


*Acknowledgements*. The authors thank Jana Meyer for particulate organic and
inorganic carbon measurements. This work was supported by the German Federal
Ministry of Education and Research (Bundesministerium für Bildung und Forschung)
in the framework of the collaborative project Biological Impacts of Ocean
Acidification (BIOACID). Kai G. Schulz is the recipient of an Australian Research
Council Future Fellowship (FT120100384). We also thank the China Postdoctoral
Science Foundation (2017M612129) and Outstanding Postdoctoral Scholarship in
State Key Laboratory of Marine Environmental Science at Xiamen University for
their supports of Yong Zhang.







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



**Figure Legends**

**Figure 1.** Optimum curve responses of measured and relative growth, particulate

organic (POC) and inorganic carbon (PIC) production rates of three *Emiliania huxleyi*

populations to a $p$CO$_2$ range from 120 µatm to 2630 µatm. Responses of measured (**a**)

and relative (**b**) growth rates to $p$CO$_2$. Responses of measured (**c**) and relative (**d**)

POC production rates to $p$CO$_2$. Responses of measured (**e**) and relative (**f**) PIC

production rates to $p$CO$_2$. Using the nonlinear regression model derived by Bach et al.

(2011), the curves were fitted based on average growth, POC and PIC production

rates of six strains from the Azores and Bergen, and of five strains from the Canary

Islands. Vertical error bars represent standard deviations of six growth, POC and PIC

production rates for the Azores and Bergen populations, and five growth, POC and

PIC production rates for the Canary Islands population. Horizontal error bars

represent standard deviations of six $p$CO$_2$ levels for the Azores and Bergen

populations and five $p$CO$_2$ levels for the Canary Islands populations. At the

population levels, 120 µatm and 2630 µatm was the lowest and highest $p$CO$_2$ level,

respectively.

**Figure 2.** Calculated optimum $p$CO$_2$, calculated maximum value and fitted relative

sensitivity constant of growth, POC and PIC production rates of each population. (**a**)

optimum $p$CO$_2$ of growth rate; (**b**) optimum $p$CO$_2$ of POC production rates; (**c**)

optimum $p$CO$_2$ of PIC production rates; (**d**) maximum growth rate, (**e**) maximum

POC production rate, (**f**) maximum PIC production rate; (**g**) relative sensitivity



constant of growth rate; **(h)** relative sensitivity constant of POC production rate; **(i)**
relative sensitivity constant of PIC production rate. The line in the middle of each box
indicates the mean of 6 or 5 optimum $p$CO$_2$, 6 or 5 maximum values, and 6 or 5
relative sensitivity constants for growth, POC and PIC production rates in each
population. Bars indicate the 99% confidence interval. The maximum or minimum
data is shown as the small line on the top or bottom of the bar, respectively. Letters in
each panel represent statistically significant differences (Tukey HSD, $p < 0.05$).

**Figure 3.** Optimum curve responses of growth, POC and PIC production rates of
individual *E. huxleyi* strains in the Azores (left), Bergen (medium) and Canary Islands
(right) populations to a CO$_2$ range from 115 µatm to 3070 µatm. Growth rates of each
strain as a function of $p$CO$_2$ within the Azores (**a**), Bergen (**b**) and Canary Islands (**c**)
populations. POC production rates of each strain as a function of $p$CO$_2$ within the
Azores (**d**), Bergen (**e**) and Canary Islands (**f**) populations. PIC production rates of
each strain as a function of $p$CO$_2$ within the Azores (**g**), Bergen (**h**) and Canary
Islands (**i**) populations. At the strain levels, 115 µatm and 3070 µatm was the lowest
and highest $p$CO$_2$ level, respectively.









**Table 1.** Surface seawater $CO_2$ levels and pH at the Azores, Bergen and Canary
Islands.

| | Location | Mean seasonal $CO_2$ (µatm) | Mean seasonal pH (total scale) | $CO_2$ variability (µatm) | References |
|---|---|---|---|---|---|
| Azores | 38°34'N, 28°42'W | 320 – 400 | 8.005 – 8.05 | 80 | Ríos et al., 2005 Wisshak et al., 2010 |
| Bergen | 60°18'N, 05°15'E | 240 – 400 | 7.98 – 8.22 | 200 | Omar et al., 2010 |
| Canary Islands | 27°58'N, 15°36'W | 320 – 400 | 8.005 – 8.05 | 80 | González-Dávila et al., 2003 |




















**Table 2.** Carbonate chemistry parameters (mean values for the beginning and end of
the incubations) of the artificial seawater for each *Emiliania huxleyi* population. pH
and TA samples were collected and measured before and at the end of incubation.
Data are expressed as mean values of six strains in the Azores and Bergen population,
and five strains in the Canary Islands population.

| | $pCO_2$ (µatm) | pH (total scale) | TA (µmol kg$^{-1}$) | DIC (µmol kg$^{-1}$) | HCO$_3^-$ (µmol kg$^{-1}$) | CO$_3^{2-}$ (µmol kg$^{-1}$) | CO$_2$ (µmol kg$^{-1}$) | $\Omega$ |
|---|---|---|---|---|---|---|---|---|
| Azores | 125±3 | 8.46±0.01 | 2358±12 | 1844±11 | 1485±13 | 355±5 | 5±0 | 8.5±0.1 |
| | 300±20 | 8.16±0.03 | 2339±27 | 2031±17 | 1803±18 | 218±13 | 11±1 | 5.2±0.3 |
| | 360±19 | 8.09±0.02 | 2322±30 | 2052±14 | 1849±9 | 190±10 | 13±1 | 4.5±0.3 |
| | 500±26 | 7.97±0.02 | 2301±23 | 2100±16 | 1933±14 | 149±8 | 18±1 | 3.5±0.2 |
| | 695±20 | 7.85±0.01 | 2317±11 | 2167±13 | 2023±14 | 118±2 | 25±1 | 2.8±0.1 |
| | 875±40 | 7.76±0.02 | 2320±19 | 2206±13 | 2076±10 | 99±5 | 32±1 | 2.4±0.1 |
| | 1110±119 | 7.66±0.05 | 2303±19 | 2222±23 | 2101±25 | 80±8 | 40±4 | 1.9±0.2 |
| | 1315±104 | 7.59±0.03 | 2308±18 | 2251±26 | 2133±26 | 70±4 | 48±4 | 1.7±0.1 |
| | 1665±107 | 7.50±0.03 | 2311±11 | 2286±15 | 2169±14 | 57±3 | 60±4 | 1.4±0.1 |
| | 1935±175 | 7.44±0.04 | 2308±15 | 2302±24 | 2183±21 | 50±4 | 70±6 | 1.2±0.1 |
| | 2490±132 | 7.33±0.02 | 2320±12 | 2350±15 | 2220±13 | 40±2 | 90±5 | 0.9±0.1 |
| Bergen | 120±3 | 8.47±0.01 | 2354±18 | 1834±18 | 1470±17 | 359±2 | 4±0 | 8.6±0.1 |
| | 290±16 | 8.17±0.02 | 2337±21 | 2024±12 | 1793±14 | 220±10 | 11±1 | 5.3±0.2 |
| | 355±18 | 8.10±0.02 | 2315±23 | 2045±11 | 1840±7 | 192±10 | 13±1 | 4.6±0.2 |
| | 490±18 | 7.98±0.02 | 2302±19 | 2096±14 | 1926±12 | 152±6 | 18±1 | 3.6±0.1 |
| | 670±22 | 7.86±0.01 | 2317±11 | 2162±10 | 2016±10 | 121±3 | 24±1 | 2.9±0.1 |
| | 855±52 | 7.77±0.03 | 2326±19 | 2206±15 | 2074±14 | 101±6 | 30±2 | 2.4±0.1 |
| | 1080±53 | 7.67±0.02 | 2316±26 | 2232±20 | 2110±18 | 83±5 | 39±2 | 2.0±0.1 |
| | 1280±71 | 7.60±0.02 | 2318±15 | 2257±17 | 2138±17 | 72±4 | 46±3 | 1.7±0.1 |
| | 1550±122 | 7.52±0.03 | 2300±19 | 2266±28 | 2150±27 | 60±4 | 56±4 | 1.4±0.1 |
| | 1800±235 | 7.47±0.05 | 2301±19 | 2286±33 | 2168±30 | 53±6 | 65±9 | 1.3±0.1 |
| | 2280±147 | 7.37±0.02 | 2309±20 | 2326±27 | 2201±24 | 42±2 | 82±5 | 1.0±0.1 |
| Canary Islands | 130±3 | 8.45±0.01 | 2344±38 | 1842±32 | 1491±26 | 347±7 | 5±0 | 8.3±0.2 |
| | 310±11 | 8.15±0.01 | 2317±24 | 2020±25 | 1798±25 | 210±4 | 11±1 | 5.0±0.1 |
| | 375±14 | 8.07±0.01 | 2295±14 | 2040±12 | 1846±13 | 182±5 | 14±1 | 4.3±0.1 |
| | 505±32 | 7.96±0.02 | 2297±19 | 2097±20 | 1930±23 | 148±7 | 18±1 | 3.5±0.2 |
| | 695±18 | 7.85±0.01 | 2312±20 | 2163±17 | 2020±15 | 118±3 | 25±1 | 2.8±0.1 |
| | 925±73 | 7.74±0.04 | 2319±26 | 2211±15 | 2083±12 | 95±8 | 33±3 | 2.3±0.1 |
| | 1180±53 | 7.64±0.02 | 2310±25 | 2239±20 | 2120±19 | 76±4 | 43±2 | 1.8±0.1 |
| | 1380±104 | 7.58±0.03 | 2323±5 | 2271±10 | 2154±11 | 68±5 | 50±4 | 1.6±0.1 |
| | 1740±98 | 7.48±0.02 | 2319±16 | 2298±16 | 2180±15 | 55±3 | 63±4 | 1.3±0.1 |
| | 2140±258 | 7.40±0.05 | 2312±9 | 2320±16 | 2197±13 | 46±5 | 78±10 | 1.1±0.1 |
| | 2630±284 | 7.31±0.04 | 2317±13 | 2363±20 | 2225±14 | 37±3 | 98±8 | 0.8±0.1 |






**Table 3.** Calculated optimum $p\mathrm{CO_2}$, calculated maximum value ($V_{\max}$) and fitted

relative sensitivity constant ($rs$, ‰) of growth, POC and PIC production rates of each

*E. huxleyi* strain.

| strain | Growth rate | | | POC production rate | | | PIC production rate | | |
|---|---|---|---|---|---|---|---|---|---|
| | optimum $p\mathrm{CO_2}$ (µatm) | $V_{\max}$ (d$^{-1}$) | $rs$ | optimum $p\mathrm{CO_2}$ (µatm) | $V_{\max}$ (pg C cell$^{-1}$ d$^{-1}$) | $rs$ | optimum $p\mathrm{CO_2}$ (µatm) | $V_{\max}$ (pg C cell$^{-1}$ d$^{-1}$) | $rs$ |
| A23 | 392 | 1.21 | 0.22 | 673 | 12.47 | 0.50 | 323 | 13.45 | 0.38 |
| A22 | 436 | 1.27 | 0.16 | 591 | 17.33 | 0.33 | 635 | 12.28 | 0.40 |
| A21 | 392 | 1.25 | 0.22 | 707 | 15.45 | 0.50 | 396 | 16.73 | 1.11 |
| A19 | 371 | 1.26 | 0.24 | 512 | 16.17 | 0.56 | 480 | 18.92 | 0.67 |
| A13 | 244 | 1.08 | 0.13 | 756 | 9.84 | 0.63 | 471 | 11.72 | 0.57 |
| A10 | 432 | 1.32 | 0.20 | 549 | 14.42 | 0.48 | 385 | 11.69 | 0.24 |
| B95 | 534 | 1.26 | 0.10 | 762 | 13.46 | 0.20 | 562 | 9.13 | 0.33 |
| B63 | 436 | 1.26 | 0.11 | 633 | 16.66 | 0.27 | 615 | 12.93 | 0.45 |
| B62 | 456 | 1.29 | 0.11 | 945 | 17.27 | 0.18 | 488 | 14.00 | 0.43 |
| B51 | 499 | 1.29 | 0.11 | 660 | 16.77 | 0.35 | 492 | 11.87 | 0.48 |
| B41 | 542 | 1.25 | 0.09 | 984 | 18.34 | 0.38 | 553 | 9.46 | 0.37 |
| B17 | 490 | 1.32 | 0.14 | 761 | 15.19 | 0.30 | 625 | 12.77 | 0.47 |
| C98 | 400 | 1.03 | 0.16 | 644 | 8.44 | 0.54 | 440 | 6.40 | 0.31 |
| C91 | 393 | 0.97 | 0.21 | 413 | 4.83 | 0.60 | 195 | 10.87 | 0.33 |
| C90 | 384 | 0.97 | 0.12 | 546 | 8.28 | 0.34 | 284 | 8.52 | 0.50 |
| C41 | 393 | 1.01 | 0.14 | 609 | 7.64 | 0.45 | 545 | 11.15 | 0.30 |
| C35 | 378 | 1.05 | 0.17 | 596 | 8.87 | 0.44 | 464 | 12.68 | 0.34 |















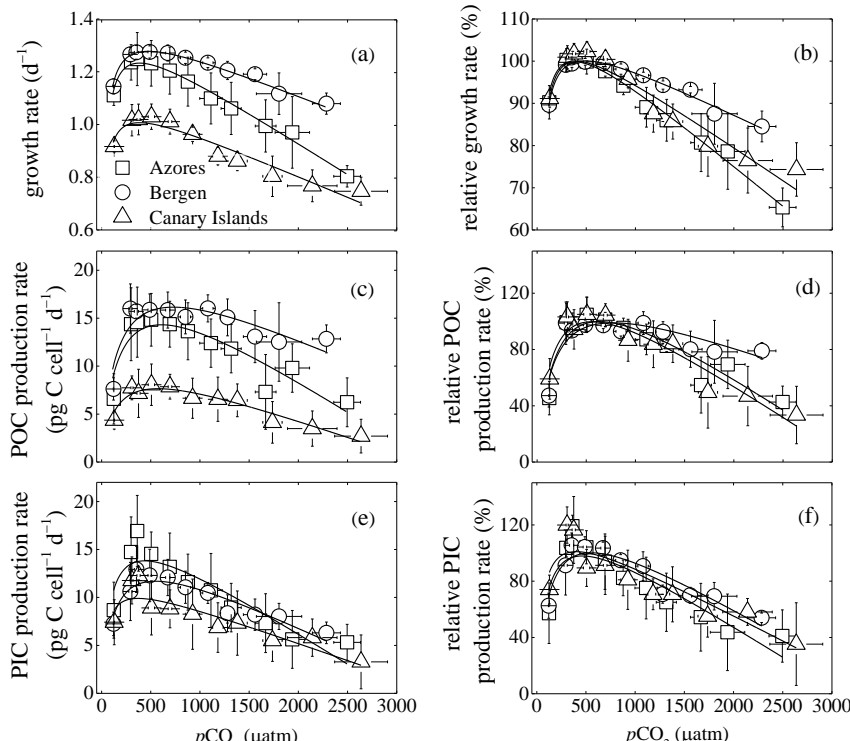





Figure 1









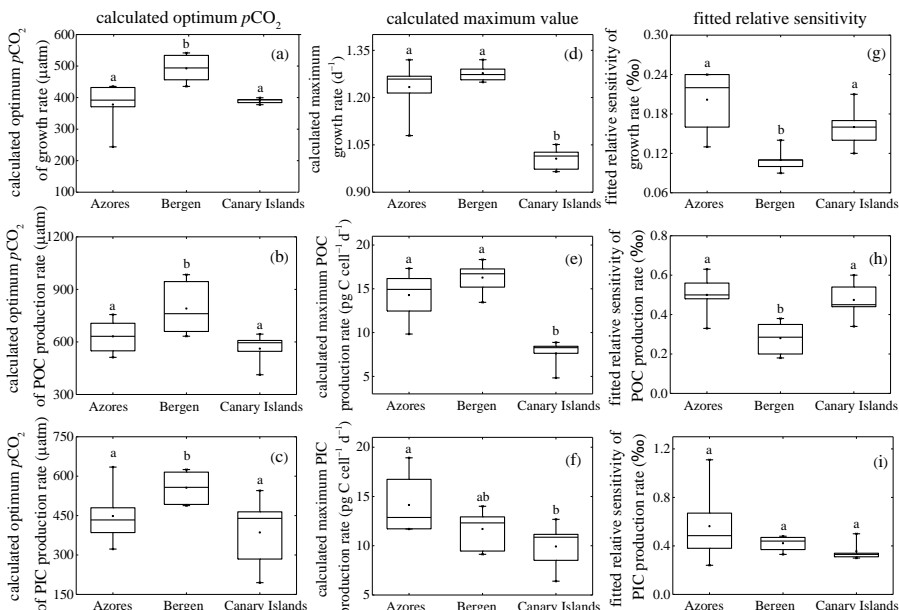





Figure 2










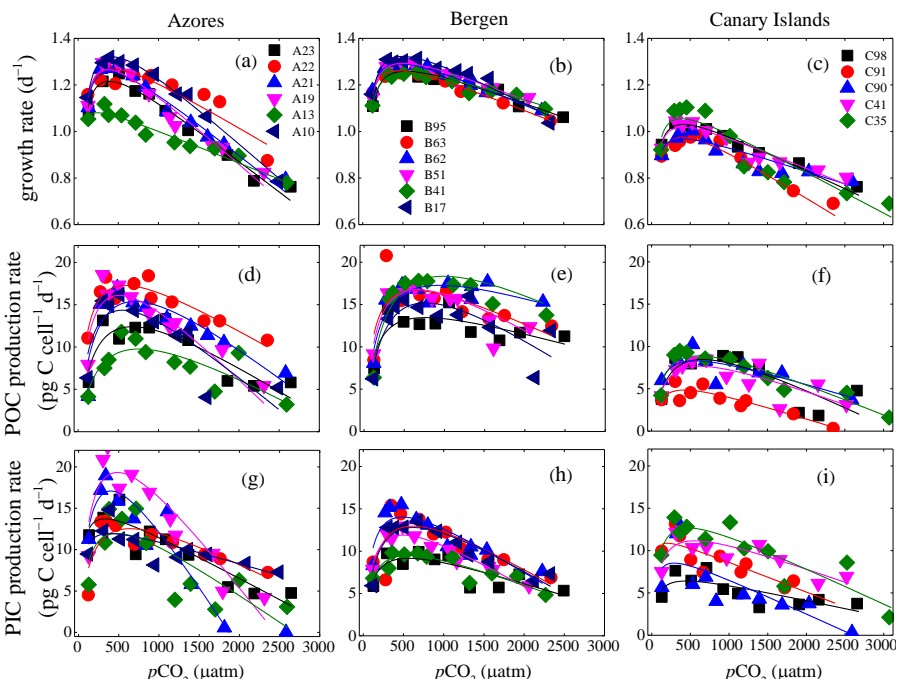





Figure 3
