# Peer review of "Population-specific responses in physiological rates of Emiliania huxleyi to a"

_Biogeosciences, 2018_

## Referee Comment (RC1) · Anonymous Referee #1 · 28 Mar 2018

Summary

Zhang et al. conducted a series of experiments with multiple strains of Emiliania huxleyi isolated from 3 different North Atlantic populations. Each strain was incubated under a broad range of pCO2 concentrations (about 120-2600uatm) but with constant total alkalinity to discern between effects due to changes in the carbonate systems and changes in CO2 levels. The physiological responses that Zhang et al measured were growth rates, PIC and POC production rates. They conclude that there were differences among strains and among populations but those differences depended on the physiological rate.

General comments

The manuscript is very well written. The ideas, methods and discussion are also clear and well structured, making the manuscript flow very well. This is high quality and thorough work and it deserves to be published. However, my main comment is perhaps related to the novelty of the work and I will make some suggestions as to how this could be addressed. Zhang et al. do a good job citing some of the previous relevant studies but their work would be better served by emphasizing how their work is significantly different and why this is important. We already know from studies like Iglesias-Rodriguez, Bach, Langer, etc., that there are CO2 effects in coccolithophore's physiological rates and we also know from Langer et al.'s work that these are species-specific and strain-specific responses, so (in my humble opinion) there is not much surprise in finding that there are population-specific differences. Throughout the manuscript the authors hint at the ideas of phenotypic plasticity and environmental variability. This, on the other hand is not so common, and I suggest that the authors elaborate more on this. They already show the pCO2 and temperature ranges in those 3 sites and it is used to explain the results. Fully accounting for this variability at the original field site is important and they should emphasize that. Acknowledging this variability is usually not done

Specific comments

While isolating the effect of CO2 from changes in TA is a great idea, it also poses the question of whether the same experiment should have been repeated letting the TA change with CO2 concentration. It begs the question of "how would the results look like if TA could change?". After all, this is a more realistic situation and it would contribute to our understanding of E hux responses to a changing World. While I acknowledge that this would be an entire new project, I think it is my role to bring it up. Perhaps acknowledging the caveat would be enough.

I am a bit confused about how the incubations were done (not saying it is wrong) but perhaps a diagram or flow chart would be helpful. I mention this in the technical comments section as well.

Also, how realistic are CO2 levels greater than 1500uatm?

It is very interesting that they found almost no differences in PIC production rates among populations, yet growth and POC production rates did show differences at the population level. Why do you think this is? One factor that the authors mention briefly is temperature, I think that temperature-adaptation and temperature-CO2 interactions might have a greater role in explaining the differences than what the authors attribute to it. In some ways the 3 populations sit along a gradient of temperature and CO2 and depending on which physiological rate is studied, one parameter might be more important than the other. Zhang et al do mention that growing certain cultures under suboptimal temperatures may have set that strain or population at a disadvantage from the beginning. Interactions between temperature and CO2 effects should not be discarded.

Another consideration is that Zhang et al do a great job by showing that there are different ranges of variability in the places where they were isolated from and they use this argument to explain the differences. However, their cultures are maintained at a constant CO2 concentration (and light pattern and temperature). As the authors suggest in this manuscript, the next generation of experiments should account for variability at its origin and hence variable environmental parameters (within a given range) in experimental designs. Plasticity and adaptation are key parameters to consider in the future.

Finally, Zhang et al found some very interesting results, some of which were not fully explored. For example, the optimum pCO2 is higher for Bergen than the other 2 regions, but the temperature optimum in Bergen is lower, what are the implications for future projections? Similarly, all strains but one showed that the pCO2 optimum for POC is greater than the optimum for PIC and growth rates, how do you think this might affect future PIC: POC ratios? What about the sensitivity constant results? OR Bergen populations experiencing the higher CO2 optimum and smallest variability between strains vs. Canary islands showing lowest optimums but highest variabilities in CO2

optimums. . .. These are just some examples of other interesting avenues to explore in the discussion.

Technical comments Line 39: than that of

Line 44-45: carbonate chemistry responses? Should it say instead "responses to changes in carbonate chemistry changes"?

Line 76: I recommend checking this new publication: Krumhardt et al. 2017. Coccolithophore growth and calcification in a changing ocean https://doi.org/10.1016/j.pocean.2017.10.007

Line 135: "consecutive incubations" and then in Line 146 "each strain was grown under 11 CO2 levels. . ." then in line 150 and 158 "at least 7 generations. . .4-7 days depending on CO2 concentration . . .". can you explain the method in more detail, I am bit confused. Perhaps a supplementary diagram or flow chart figure would help.

Line 202: For Eq 4 and 5, you cited Bach et al 2011, but could you please elaborate on this method. Can you also explain the sensitivity constant a bit more?

Line 207: Do these refer to figure S3?

Line 295: "These findings indicate that the Bergen population may be more tolerant . . ." This is a great result! Environmental variability can tell us something about phenotypic plasticity.

Line 323 "likely causes the lower the carbon. . ." consider moving "the"

Line 343: add and "s" to proton

Line 345: consider adding "and" before "corresponding"

Line 352: this conclusion seems to be out of place and not well justified

Lines 334-372: some very interesting ideas here but these paragraphs needs some tightening.

Line 367-369: do you mean "dominated" or "dominating"? not sure I follow this argument.

---

## Referee Comment (RC2) · Anonymous Referee #2 · 28 Mar 2018

GENERAL COMMENTS

The paper by Zhang et al. presents results from a large number of experiments on multiple geographically distinct strains of the coccolithophore Emiliania huxleyi. Each strain was exposed to a wide range of pCO2 concentrations and the authors examined differences in growth rates, photosynthetic rates (POC production) and calcification rates (PIC production). The authors conclude that significant variability exists in population-level sensitivity of physiological rates (most clearly growth and POC production) to pCO2. The paper is well written, with the data supporting the conclusions and the authors make some important and insightful conclusions. I have only two minor comments.

The first comment relates to a lack of any discussion or presentation of the variability

in PIC:POC ratios and POC (or PIC) production between the different strains. Further information on the level of inter-strain variability in these parameters would strengthen and support the wider implications and conclusions made in the discussion. The second comment relates to the authors consideration of variability and stability in the different environmental conditions of the strain isolation locations – a large factor in these differences is likely to relate to different seasonal cycles and environmental drivers (ice-melt, riverine input, upwelling, etc). However, the authors only hint at the different factors influencing the relative stability of the different locations. Large-scale environmental differences will directly relate to the stability of the environment, as well as differing potential future perturbations for each of them. Again, making these differences more explicit would support the wider implications of the study.

SPECIFIC COMMENTS

Ln 27: Clarity is needed in the abstract on what the authors mean in terms of population-specific responses.

Ln 28: More information on number of strains per environment would be good in the abstract.

Ln 32: 'expected optimum curve responses' – may be expected by authors but not clear in the abstract. Some further background would be good.

Ln 37: Could the authors elaborate more in terms of the role of seasonality (or lack thereof) in the stability of oceanic conditions.

Lns 91-92: Would the authors consider adding 'geographically-distinct' strains to this line to emphasize both the importance of their own insights and the more general need to consider different strains of other widespread species.

Lns 103-104: A plastic response also allows a strain to acclimate across an environmental gradient and widen its bio-geographical distribution. Rather than focus on just environmental change, what about environmental variability.

Ln 126: How were all strains characterized and confirmed to be morphotype A (i.e. Distal shield length? Central area characteristics?)?

Ln 140-141: Is this statement ('the best compromise') appropriate based on the authors end conclusion that the low experiment temperature relative to optimum growth conditions for the Canary Islands strains led to their low growth (and POC production)? It seems to be a compromise that had a definitive influence on the end outcome of the experiments. Is it not simpler to just delete this section (from the point of 'which ..' to the end) and come back to this in the discussion?

Lns 152-153 (cf Lns 174-175): How were initial cell densities measured/estimated?

Lns 289-290: An important result that should be emphasized in the abstract and conclusions.

Lns 322-324: Suggest deleting 'causes' from this sentence.

Ln 351-352: Another potentially important conclusion, especially given the emphasis on determining time-dependent (or space-dependent) variations in coccolith-specific PIC quotas. However, the current paper lacks any details of the strain-specific variability in PIC quota and to what extent the different trends in $pCO_2$-sensitivity (e.g. Fig. 3e) are driven by changes in growth rate and/or cellular (or coccolith) specific PIC quota. Can strain-specific information on PIC quota be added to the supplementary material to support this point with experimental data?

Ln 374: A two line conclusion seems relatively short based on the significant statements made in the conclusions. Either expand or delete?

---

## Author Response (AR1)

**Responses to comments**

Dear referees,

We thank you for your supportive comments on our manuscript. Our detailed response in blue text to your comments is attached. Changes to the manuscript text are underlined.

**Responses to comments of referee 1**

Summary

Zhang et al. conducted a series of experiments with multiple strains of Emiliania huxleyi isolated from 3 different North Atlantic populations. Each strain was incubated under a broad range of pCO2 concentrations (about 120-2600uatm) but with constant total alkalinity to discern between effects due to changes in the carbonate systems and changes in CO2 levels. The physiological responses that Zhang et al measured were growth rates, PIC and POC production rates. They conclude that there were differences among strains and among populations but those differences depended on the physiological rate.

General comments

The manuscript is very well written. The ideas, methods and discussion are also clear and well structured, making the manuscript flow very well. This is high quality and thorough work and it deserves to be published. However, my main comment is perhaps related to the novelty of the work and I will make some suggestions as to how this could be addressed. Zhang et al. do a good job citing some of the previous relevant studies but their work would be better served by emphasizing how their work is significantly different and why this is important. We already know from studies like Iglesias-Rodriguez, Bach, Langer, etc., that there are CO2 effects in coccolithophore's physiological rates and we also know from Langer et al.'s work that these are species-specific and strain-specific responses, so (in my humble opinion) there is not much surprise in finding that there are population-specific differences. Throughout the manuscript the authors hint at the ideas of phenotypic plasticity and environmental variability. This, on the other hand is not so common, and I suggest that the authors elaborate more on this. They already show the pCO2 and temperature ranges in those 3 sites and it is used to explain the results. Fully accounting for this variability at the original field site is important and they should emphasize that. Acknowledging this variability is usually not done。

Response: We thank this referee for the positive comments. We summarized responses of growth, POC and PIC production rates of different *Emiliania huxleyi* strains to $CO_2$ and found that most of these studies focused on a few strains or a narrow range of $CO_2$ level (Table R1). In this study, we used 17 strains and measured growth, POC and PIC production rates at 120 µatm to 2630 µatm, which are different from previous studies. **These contents were shown in lines 84–87.**

When exposed to 16 ºC, growth rate of the Canary Islands population might have been already below their optimum and hence significantly reduced in comparison to the other populations  (Fig. 2d). These changes are in **lines 333–336.**

Phenotypic plasticity constitutes an advantage for individual strains to acclimate and adapt to elevated $p$CO$_2$ by changing  fitness-relevant traits and potentially to attenuate the effects of changing environments on fitness-relevant traits (Schaum et al., 2013). These changes are in **lines 388–391.**

Physiological variability makes a population more resilient.  increases its ability to persist in variable environments and potentially forms the basis for selection (Gsell et al., 2012; Hattich et al., 2017). These changes are in **lines 395–397**.

Table R1. Summary of the physiological responses of different *E. huxleyi* genotypes to various $p$CO$_2$ ranges at constant alkalinity condition. Symbols indicate: ↑ increased response, — no response, ↓ decreased response, ∩ optimum response.

| *E. huxleyi* genotype | Isolated site | $p$CO$_2$ range (µatm) | Growth rate | POC pro. | PIC pro. | Incubation temp. (ºC) | Reference |
|---|---|---|---|---|---|---|---|
| AC472 | South Pacific, New Zealand | 400 to 760 | ↑ | — | ↑ | 19 | Fiorini et al., (2011) |
| EHTB 11.15 | Trumpeter Bay, Tasmania | 375 to 1650 | — | ∩ | ↓ | 14 | Müller et al., (2015) |
| EHSO 5.14 | Southern Ocean | 300 to 1680 | ↓ | ∩ | ∩ | 14 | Müller et al., (2015) |
| EHSO 5.11 | Southern Ocean | 259 to 1255 | ∩ | ∩ | ∩ | 14 | Müller et al., (2015) |
| NIWA1108 | Chatham Rise, New Zealand | 80 to 1080 | ↑ | ↑ | ∩ | 4-25 | Feng et al., (2017) |
| PLY M219 (NZEH) | New Zealand | 380 to 750 | ↓ | ↓ | ↓ | 20 | Shi et al., (2009) |
| PLY M219 (NZEH) | New Zealand | 404 to 1066 | ↓ | ↑ | ↓ | 15 | Hoppe et al., (2011) |
| PML B92/11A | Bergen, Norway | 152 to 885 | — | ↑ | ↓ | 15 | Riebesell et al., (2000) |
| PML B92/11A | Bergen, Norway | 20 to 6000 | ∩ | ∩ | ∩ | 15 | Bach et al., (2011) |
| RCC1212 | South Atlantic, off South Africa | 194 to 1096 | ↓ | ∩ | ↓ | 20 | Langer et al., (2009) |
| RCC1216 | Tasman Sea, off New Zealand | 218 to 1201 | ↓ | ↑ | ↓ | 17 | Langer et al., (2009) |
| RCC1238 | North Atlantic, off Japan | 206 to 929 | ↑ | ∩ | — | 20 | Langer et al., (2009) |
| RCC1256 | North Atlantic, off Iceland | 193 to 915 | ↓ | ∩ | ∩ | 17 | Langer et al., (2009) |

| RCC1256 | Iceland | 191 to 846 | ↓ | ↓ | ↓ | 15 | Hoppe et al., (2011) |
|---|---|---|---|---|---|---|---|
| NZEH | New Zealand | 280 to 750 | ↓ | ↑ | ↑ | 19 | Iglesias-Rodriguez et al., (2008) |
| NZEH | New Zealand | 395 to 1340 | ↓ | ↑ | ↑ | 19 | Jones et al., (2013) |

Bach, L. T., Riebesell, U., and Schulz, K.G.: Distinguishing between the effects of ocean acidification and ocean carbonation in the coccolithophore *Emiliania huxleyi*, Limnol. Oceanogr., 56, 2040–2050, 2011.

Beardall, J., and Raven, J. A.: Potential effects of global change on microalgal photosynthesis, growth and ecology, Phycologia, 43, 26–40, 2004.

Feng, Y. Y., Roleda, M. Y., Armstrong, E., Boyd, P. W., and Hurd, C. L.: Environmental controls on the growth, photosynthetic and calcification rates of a Southern Hemisphere strain of the coccolithophore *Emiliania huxleyi*, Limnol. Oceanogr., 62, 519–540.

Fiorini, S., Middelburg, J. J., and Gattuso, J. P.: Testing the effects of elevated $pCO_2$ on coccolithophores (prymnesiophyceae): comparison between haploid and diploid life stages, J. Phycol., 47, 1281–1291, 2011.

Hoppe, C. J. M., Langer, G., and Rost, B.: *Emiliania huxleyi* shows identical responses to elevated $pCO_2$ in TA and DIC manipulations, J. Exp. Mar. Biol. Ecol., 406, 54–62, 2011.

Iglesias-Rodriguez, M. D., Halloran, P. R., Rickaby, R. E. M., Hall, I. R., Colmenero-Hidalgo, E., Gittins, J. R., Green, D. R. H., Tyrrell, T., Gibbs, S. J., von Dassow, P., Rehm, E., Armbrust, E.V., and Boessenkool, K.P.: Phytoplankton calcification in a high-$CO_2$ world, Science, 320, 336–340, 2008.

Jones, B. M., Iglesias-Rodriguez, M. D., Skipp, P. J., Edwards, R. J., Greaves, M. J., Young, J. R., Elderfield, H., and O'Connor, C. D.: Responses of the *Emiliania huxleyi* proteome to ocean acidification, PLoS One 8(4), e61868, doi: 10.1371/journal.pone.0061868, 2013.

Langer, G., Nehrke, G., Probert, I., Ly, J., and Ziveri, P.: Strain-specific responses of *Emiliania huxleyi* to changing seawater carbonate chemistry, Biogeosciences, 6, 2637–2646, 2009.

Müller, M. N., Trull, T. W., and Hallegraeff, G. M.: Differing responses of three Southern Ocean *Emiliania huxleyi* ecotypes to changing seawater carbonate chemistry, Mar. Ecol. Prog. Ser., 531, 81–90, 2015.

Riebesell, U., Zondervan, I., Rost, B., Tortell, P. D., Zeebe, R. E., and Morel, F. M. M.: Reduced calcification of marine plankton in response to increased atmospheric $CO_2$, Nature, 407, 364–367, 2000.

Shi, D., Xu, Y., and Morel, F. M. M.: Effects of the pH/$pCO_2$ control method on medium chemistry and phytoplankton growth, Biogeosciences, 6, 1199–1207, 2009.

Specific comments

While isolating the effect of CO2 from changes in TA is a great idea, it also poses the question of whether the same experiment should have been repeated letting the TA change with CO2 concentration. It begs the question of "how would the results look like if TA could change?". After all, this is a more realistic situation and it would contribute to our understanding of E hux responses to a changing World. While I acknowledge that this would be an entire new project, I think it is my role to bring it up. Perhaps acknowledging the caveat would be enough.

Response: We did not 'isolate the effect of $CO_2$ from changes in TA', and our $CO_2$ manipulations are mimicking ongoing ocean acidification where $CO_2$/pH and DIC changes at constant TA.

As shown in Tables R2 and R3, rising $pCO_2$ level dominantly decreased pH at increasing TA conditions. According to studies of Bach et al. (2011), after optimum $CO_2$ levels, low pH inhibited growth, POC and PIC production. Thus, we expected that growth, POC and PIC production rates should show optimal curve responses to a broad $CO_2$ range at changing TA.

Table R2. Carbonate chemistry parameter at constant $pCO_2$ levels.

| TA ($\mu mol\ L^{-1}$) | DIC ($\mu mol\ kg^{-1}$) | pH (total scale) | $pCO_2$ ($\mu atm$) | $HCO_3^-$ ($\mu mol\ kg^{-1}$) | $CO_3^{2-}$ ($\mu mol\ kg^{-1}$) | $CO_2$ ($\mu mol\ kg^{-1}$) | $\Omega$ |
|---|---|---|---|---|---|---|---|
| 1500 | 1351.6 | 7.887 | 400 | 1245.0 | 93.7 | 12.9 | 2.24 |
| 1600 | 1436.9 | 7.912 | 400 | 1318.8 | 105.1 | 12.9 | 2.51 |
| 1700 | 1521.8 | 7.935 | 400 | 1391.8 | 117.1 | 12.9 | 2.80 |
| 1800 | 1606.3 | 7.957 | 400 | 1463.9 | 129.5 | 12.9 | 3.10 |
| 1900 | 1690.4 | 7.978 | 400 | 1535.1 | 142.4 | 12.9 | 3.41 |
| 2000 | 1774.2 | 7.997 | 400 | 1605.4 | 155.8 | 12.9 | 3.73 |
| 2100 | 1857.5 | 8.016 | 400 | 1675.0 | 169.6 | 12.9 | 4.06 |
| 2200 | 1940.6 | 8.033 | 400 | 1743.8 | 183.8 | 12.9 | 4.40 |
| 2300 | 2023.3 | 8.050 | 400 | 1811.9 | 198.4 | 12.9 | 4.75 |
| 2400 | 2105.6 | 8.066 | 400 | 1879.2 | 213.5 | 12.9 | 5.11 |
| 2500 | 2187.7 | 8.081 | 400 | 1945.9 | 228.9 | 12.9 | 5.47 |
| 2600 | 2269.4 | 8.095 | 400 | 2011.8 | 244.7 | 12.9 | 5.85 |
| 2700 | 2350.8 | 8.109 | 400 | 2077.1 | 260.8 | 12.9 | 6.24 |
| 2800 | 2432.0 | 8.122 | 400 | 2141.8 | 277.3 | 12.9 | 6.63 |

Table R3. Carbonate chemistry parameter at changing $pCO_2$ levels and changing TA conditions.

| TA ($\mu mol\ L^{-1}$) | DIC ($\mu mol\ kg^{-1}$) | pH (total scale) | $pCO_2$ ($\mu atm$) | $HCO_3^-$ ($\mu mol\ kg^{-1}$) | $CO_3^{2-}$ ($\mu mol\ kg^{-1}$) | $CO_2$ ($\mu mol\ kg^{-1}$) | $\Omega$ |
|---|---|---|---|---|---|---|---|
| 1500 | 1254.0 | 8.134 | 200 | 1101.0 | 146.5 | 6.5 | 3.51 |
| 1600 | 1436.9 | 7.912 | 400 | 1318.8 | 105.1 | 12.9 | 2.51 |
| 1700 | 1576.7 | 7.783 | 600 | 1470.2 | 87.1 | 19.4 | 2.08 |
| 1800 | 1701.7 | 7.694 | 800 | 1598.6 | 77.2 | 25.8 | 1.85 |
| 1900 | 1819.7 | 7.628 | 1000 | 1716.2 | 71.2 | 32.3 | 1.70 |
| 2000 | 1934.0 | 7.576 | 1200 | 1827.9 | 67.3 | 38.8 | 1.61 |
| 2100 | 2046.0 | 7.534 | 1400 | 1936.0 | 64.7 | 45.2 | 1.55 |
| 2200 | 2156.4 | 7.500 | 1600 | 2041.7 | 63.0 | 51.7 | 1.51 |
| 2300 | 2265.8 | 7.470 | 1800 | 2145.8 | 61.8 | 58.1 | 1.48 |
| 2400 | 2374.4 | 7.445 | 2000 | 2248.7 | 61.1 | 64.6 | 1.46 |
| 2500 | 2482.4 | 7.422 | 2200 | 2350.7 | 60.7 | 71.1 | 1.45 |

| 2600 | 2590.1 | 7.403 | 2400 | 2452.0 | 60.6 | 77.5 | 1.45 |
|------|--------|-------|------|--------|------|------|------|
| 2700 | 2697.4 | 7.386 | 2600 | 2552.8 | 60.6 | 84.0 | 1.45 |
| 2800 | 2804.4 | 7.370 | 2800 | 2653.2 | 60.8 | 90.4 | 1.45 |

I am a bit confused about how the incubations were done (not saying it is wrong) but perhaps a diagram or flow chart would be helpful. I mention this in the technical comments section as well.

Response: We agree with this referee and present a flow chart which shows the experimental protocol. This flow chart was added in the supplement information as **Figure S1**.

[Figure]

**Figure R1 (S1).** A flow chart of the experimental protocol.

Also, how realistic are $CO_2$ levels greater than 1500 uatm?

Response: According to business-as-usual $CO_2$ emissions (RCP8.5), atmospheric $CO_2$ level are projected higher than 1500 ppmv after 2200 (Meinshausen et al. 2011).

Meinshausen, M., Smith, S. J., Calvin, K., Daniel, J. S., Kainuma, M. L. T., Lamarque, J., Matsumoto, K., Montzka, S. A., Raper, S. C. B., Riahi, K., Thomson, A., Velders, G. J. M., and van Vuuren, D. P. P.: The RCP greenhouse gas concentrations and their extensions from 1765 to 2300, Climatic Change, 109, 213–241, 2011.

It is very interesting that they found almost no differences in PIC production rates among populations, yet growth and POC production rates did show differences at the population level. Why do you think this is? One factor that the authors mention briefly is temperature, I think that temperature-adaptation and temperature-CO2 interactions might have a greater role in explaining the differences than what the authors attribute to it. In some ways the 3 populations sit along a gradient of temperature and CO2 and depending on which physiological rate is studied, one parameter might be more important than the other. Zhang et al do mention that growing certain cultures under suboptimal temperatures may have set that strain or population at a disadvantage from the beginning. Interactions between temperature and CO2 effects should not be discarded.

Response: We thank the referee for this suggestion.

These contents 'One of the reasons may be that compared to the Azores and Bergen populations, 16 $^{\circ}C$ likely causes lower carbon uptake and carbon-use efficiency of the Canary Islands population (Sett et al., 2014).' were replaced by 'Furthermore, compared to the Canary Islands population, the Azores population had higher maximum growth and POC production rates, and similar optimum $CO_2$ for these physiological rates. Again, this might be related to sub-optimal incubation conditions as temperature has been found to significantly modulate $CO_2$ responses in coccolithophores in terms of maximum rates, $CO_2$ optima and half-saturation, and $H^+$ sensitivity (De Bodt et al., 2010; Sett et al., 2014; Gafar et al., 2018; Gafar and Schulz, 2018). In a similar fashion light can also modulate $CO_2$ responses, hence different requirements by strains adapted to different light availabilities could also explain our observations (Zhang et al., 2015; Gafar et al., 2018; Gafar and Schulz, 2018).' These changes are in **lines 337–348.**

In addition, the Canary Islands population showed smallest variability in optimum $pCO_2$ and maximum values for growth and POC production rates (Fig. 2). The reason may be that low incubation temperature predominantly limited growth and POC production rates of the Canary Islands population, and decreased the sensitivies of these physiological rates to rising $pCO_2$. These changes are in **lines 350–355**

De Bodt, C., Van Oostende, N., Harlay, J., Sabbe, K., and Chou, L: Individual and interacting effects of $pCO_2$ and temperature on *Emiliania huxleyi* calcification: study of the calcite production, the coccolith morphology and the coccosphere size, Biogeosciences, 1401–1412, 2010.

Gafar, N. A., Eyre, B. D., and Schulz, K. G. : A conceptual model for projecting coccolithophorid growth, calcification and photosynthetic carbon fixation rates in response to global ocean change, Front. Mar. Sci., 4, 433, doi: 10.3389/fmars.2017.00433, 2018.

Gafar, N. A., and Schulz, K. G. : A niche comparison of *Emiliania huxleyi* and *Gephyrocapsa oceanica* and potential effects of climate change, Biogeosci. Discuss., doi: 10.5194/bg-2018-88.

Another consideration is that Zhang et al do a great job by showing that there are different ranges of variability in the places where they were isolated from and they use this argument to explain the differences. However, their cultures are maintained at a constant CO2 concentration (and light pattern and temperature). As the authors suggest in this manuscript, the next generation of experiments should account for variability at its origin and hence variable environmental parameters (within a given range) in experimental designs. Plasticity and adaptation are key parameters to consider in the future.

Response: we agree with this referee.

Phenotypic plasticity constitutes an advantage for individual strains to acclimate and adapt to elevated $p$CO$_2$ by changing fitness-relevant traits and potentially to attenuate the short-term effects of changing environments on fitness-relevant traits (Schaum et al., 2013). These changes are in **lines 388–391.**

Finally, Zhang et al found some very interesting results, some of which were not fully explored. For example, the optimum pCO2 is higher for Bergen than the other 2 regions, but the temperature optimum in Bergen is lower, what are the implications for future projections? Similarly, all strains but one showed that the pCO2 optimum for POC is greater than the optimum for PIC and growth rates, how do you think this might affect future PIC: POC ratios? What about the sensitivity constant results? OR Bergen populations experiencing the higher CO2 optimum and smallest variability between strains vs. Canary islands showing lowest optimums but highest variabilities in CO2 optimums….. These are just some examples of other interesting avenues to explore in the discussion.

Response: Agreed. The optimum temperature for growth of the Bergen population was about 22 °C and was 5 °C higher than the maximum SST in Bergen waters (Zhang et al. 2014). Furthermore, in comparison to the Azores and Canary Islands populations, larger optimum $p$CO$_2$ of growth rate indicates that the Bergen population may benefit more from the rising CO$_2$ levels at increasing temperatures. **These contents were added in lines 367–372.**

As shown in Fig. R2 (or S6 in the supplement), PIC : POC ratios of the Azores and Bergen populations declined with rising $p$CO$_2$, whereas PIC : POC ratios of the Canary Islands population were rather constant (Fig. S6). As changes in PIC : POC ratios of coccolithophore blooms may impact on the biological carbon pump, different regions might see different changes in the future ocean. **These contents underlined were added in lines 372–376.**

In the manuscript or in Fig. 2, low sensitivity constant of growth rate of the Bergen population corresponded to high optimum CO$_2$ level. These contents were shown in **lines 304–306.**

In addition, the Canary Islands population showed smallest variability in optimum $p$CO$_2$ and maximum values for growth and POC production rates (Fig. 2). The reason may be that low incubation temperature predominantly limited growth and POC production rates of Canary Islands population, and decreased the sensitivities of these physiological rates to rising $p$CO$_2$. These changes are in **lines 350–355.**

[Figure]

**Figure R2 (S6).** Responses of PIC : POC ratio of the Azores (square), Bergen (circular) and Canary Islands (diamond) populations to a CO$_2$ range from 120 µatm to 2630 µatm.

[Figure]

**Figure R3 (S7).** Response of PIC : POC ratio of individual *E. huxleyi* strain in the Azores (A), Bergen (B) and Canary Islands (C) populations to a $CO_2$ range from 115 µatm to 3070 µatm.

Technical comments Line 39: than that of

Response: Maximum growth and POC production rates of the Azores and Bergen populations were similar and significantly higher than that of the Canary Islands population. This change is in **line 41**.

Line 44-45: carbonate chemistry responses? Should it say instead "responses to changes in carbonate chemistry changes"?

Response: The existence of distinct  responses to changes in carbonate chemistry between and within populations will likely benefit *E. huxleyi* to acclimate and adapt to rising $CO_2$ levels in the oceans. These changes are in **lines 48–51.**

Line 76: I recommend checking this new publication: Krumhardt et al. 2017. Coccolithophore growth and calcification in a changing ocean https://doi.org/10.1016/j.pocean.2017.10.007

Response: Krumhardt et al. (2017) developed an empirical coccolithophore model to investigate responses of growth and calcification of coccolithophores to changing environments (temperature,

CO$_2$, nutrient concentrations). This paper is now cited on **line 76**.

The coccolithophore *Emiliania huxleyi* forms extensive blooms under favourable light intensity, temperature and nutrient conditions, with different morphotypes in certain regions (Cook et al., 2011; Henderiks et al., 2012; Smith et al., 2012; Balch et al., 2014; Krumhardt et al., 2017). This change is in **Line 76.**

Krumhardt, K. M., Lovenduski, N. S., Debora Iglesias-Rodriguez, M., and Kleypas, J. A.: Coccolithophore growth and calcification in a changing ocean, Prog. Oceanogr., 159, 276–295.

Line 135: "consecutive incubations" and then in Line 146 "each strain was grown under 11 CO2 levels: : :" then in line 150 and 158 "at least 7 generations: : :4-7 days depending on CO2 concentration : : :". can you explain the method in more detail, I am bit confused. Perhaps a supplementary diagram or flow chart figure would help.

Response: As mentioned above, a flow chart showing the experimental protocol was added to the supplement information (Figure S1).

The experiment was performed in six consecutive incubations, with one strain from each population (Azores, Bergen, Canary Islands) being cultured at a time (Fig. S1). This change is in lines **139–140.**

Line 202: For Eq 4 and 5, you cited Bach et al 2011, but could you please elaborate on this method. Can you also explain the sensitivity constant a bit more?

Response: In a broad $p$CO$_2$ range, physiological rates are expected to initially increase quickly until reaching an optimum and then decline towards further increasing CO$_2$ levels (e.g. Krug et al. 2011). Hence we used the following modified Michaelis-Menten equation (Bach et al. 2011) which was fitted to measured cellular growth, POC and PIC production rates and yield theoretical optimum $p$CO$_2$ and maximum values for each of the three populations (combining the data of five or six strains) (Bach et al., 2011). These changes are in **lines 202–209.**

$s$, the sensitivity constant, depicts the slope of the decline after optimum CO$_2$ levels in response to rising H$^+$. These changes are in **lines 211–212.**

Line 207: Do these refer to figure S3?

Response: This refers to fig. 1 and fig. S2 (**Lines 218–220**) Relative growth, POC and PIC production rates of each population are shown in Fig. 1b,d,f. Relative POC and PIC quotas of each population were shown in Fig. S2.

Line 295: "These findings indicate that the Bergen population may be more tolerant….." This is a great result! Environmental variability can tell us something about phenotypic plasticity.

Response: (**Lines 315–317**) Large environmental variability usually results in high tolerance of phytoplankton (Doblin and van Sebille, 2016). In this study, we cannot say that large environmental variability result in large or low phenotypic plasticity.

Doblin, M. A., and van Sebille, E.: Drift in ocean currents impacts intergenerational microbial exposure to temperature, Proc. Natl. Acad. Sci. USA., 113, 5700–5705, doi: 10.1073/pnas.1521093113, 2016.

Line 323 "likely causes the lower the carbon: : :" consider moving "the"
Response: We delete '*One of the reasons may be that compared to the Azores and Bergen populations, 16 ºC likely causes lower the carbon uptake and carbon use efficiency of the Canary Islands population*' in **lines 345–348.**

Line 343: add and "s" to proton

**Response:** In addition, *E. huxleyi* is thought to utilize $HCO_3^-$ for calcification which generates protons, and increase in proton concentration may mitigate the potential of the ocean to absorb atmospheric $CO_2$ (Paasche, 2002). These changes are in **lines 379–382.**

Line 345: consider adding "and" before "corresponding"
Response: We deleted this sentence '*Thus, due to population-specific growth and PIC production rates or quotas, changes in species composition, corresponding changes in PIC productions, may affect the ability of the ocean to take up $CO_2$.*' in **lines 382–385**.

Line 352: this conclusion seems to be out of place and not well justified
**Response:** We deleted this sentence '*Additionally, our results also suggest that strain-specific PIC quota may be the basis of variation in coccoliths of E. huxleyi within the morphotype A (Fig. S4) (Young, 1994; Paasche, 2002).*' in **lines 391–393.**

Lines 334-372: some very interesting ideas here but these paragraphs need some tightening.
**Response:** According to suggestions of this referee, we added and deleted some contents: The ability to adapt to diverse environmental conditions is supposed to be one reason for the global distribution of *E. huxleyi* (Paasche, 2002), spanning a temperature range of about 30 ºC. The optimum temperature for growth of the Bergen population was about 22 ºC and was 5 ºC higher than the maximum SST in Bergen waters (Zhang et al. 2014). Furthermore, in comparison to the Azores and Canary Islands populations, larger optimum *p*$CO_2$ of growth rate indicates that the Bergen population may benefit more from the rising $CO_2$ levels at increasing temperatures. PIC : POC ratios of the Azores and Bergen populations declined with rising *p*$CO_2$, whereas PIC : POC ratios of the Canary Islands population were rather constant (Fig. S6). As changes in PIC :POC ratios of coccolithophore blooms may impact on the biological carbon pump, different regions might see different changes in the future ocean. In natural seawater, due to ocean currents and gene flow, populations at any given location may get replaced by immigrant genotypes transported there from other locations (Doblin and van Sebille, 2016). In addition, *E. huxleyi* is thought to utilize $HCO_3^-$ for calcification which generates protons, and increase in proton concentration may mitigate the potential of the ocean to absorb atmospheric $CO_2$ (Paasche, 2002). These changes are in **lines 365–382.**

Line 367-369: do you mean "dominated" or "dominating"? not sure I follow this argument.

**Response:** Further, a significant positive correlation between growth and POC production rate or POC quota (Fig. S5) indicates that the dominating strains will also take up or fix dissolved inorganic carbon faster. These changes are in **lines 408–411.**

**Responses to comments of referee 2 are shown as following:**

GENERAL COMMENTS

The paper by Zhang et al. presents results from a large number of experiments on multiple geographically distinct strains of the coccolithophore Emiliania huxleyi. Each strain was exposed to a wide range of pCO2 concentrations and the authors examined differences in growth rates, photosynthetic rates (POC production) and calcification rates (PIC production). The authors conclude that significant variability exists in population-level sensitivity of physiological rates (most clearly growth and POC production) to pCO2. The paper is well written, with the data supporting the conclusions and the authors make some important and insightful conclusions. I have only two minor comments.

The first comment relates to a lack of any discussion or presentation of the variability in PIC:POC ratios and POC (or PIC) production between the different strains. Further information on the level of inter-strain variability in these parameters would strengthen and support the wider implications and conclusions made in the discussion. The second comment relates to the authors consideration of variability and stability in the different environmental conditions of the strain isolation locations – a large factor in these differences is likely to relate to different seasonal cycles and environmental drivers (ice-melt, riverine input, upwelling, etc). However, the authors only hint at the different factors influencing the relative stability of the different locations. Large-scale environmental differences will directly relate to the stability of the environment, as well as differing potential future perturbations for each of them. Again, making these differences more explicit would support the wider implications of the study.

Response: We cultured 17 *Emiliania huxleyi* strains at 11 $p$CO$_2$ levels with no replicate. At each $p$CO$_2$ level, there is no replicate and this is the main reason that we did not discuss variability in physiological rates between strains within the population.

Regarding the variability in the PIC : POC ratio between the populations, we added these contents 'PIC : POC ratios of the Azores and Bergen populations declined with rising $p$CO$_2$, whereas PIC : POC ratios of the Canary Islands population were rather constant (Fig. S6). As changes in PIC : POC ratios of coccolithophore blooms may impact on the biological carbon pump, different regions might see different changes in the future ocean.' **in lines 372–376.**

We added these contents: 'In addition, due to riverine input, seawater upwelling and metabolic activity of plankton communities, environmental variability in coastal water are larger than in open-ocean ecosystems (Duarte and Cerbrian, 1996).' in **lines 313–315.**

Duarte, C. M., and Cerbrian, J.: The fate of marine autotrophic production, Limnol. Oceanogr., 41, 1758–1766, 1996.

SPECIFIC COMMENTS

Ln 27: Clarity is needed in the abstract on what the authors mean in terms of population-specific responses.

Response: In this study, 'population-specific responses' mean that growth, POC and PIC production rates of three *Emiliania huxleyi* populations were different at the same incubation conditions.

In the present study, we investigated the  specific responses of growth, particulate organic (POC) and inorganic carbon (PIC) production rates of  3 populations of the coccolithophore *Emiliania huxleyi* from three regions in the North Atlantic Ocean (Azores: 6 strains, Canary Islands: 5 strains, and Norwegian coast near Bergen: 6 strains) to a $CO_2$ partial pressure ($pCO_2$) range from 120 µatm to 2630 µatm. These changes are in **lines 27–32.**

Ln 28: More information on number of strains per environment would be good in the abstract.
Response: **For lines 27–32: see above**.

Ln 32: 'expected optimum curve responses' – may be expected by authors but not clear in the abstract. Some further background would be good.
Response: Physiological rates of each population and individual strain increased with rising $pCO_2$ levels, reached maximum and declined thereafter. These changes are in **lines 32–34.**

Ln 37: Could the authors elaborate more in terms of the role of seasonality (or lack thereof) in the stability of oceanic conditions.
Response: This may be due to the large environmental variability including large $pCO_2$ and pH fluctuations in coastal waters off Bergen compared to the rather stable oceanic conditions at the other two sites. These changes are in **lines 37–39**.

In the discussion section: we added this sentence 'In addition, due to riverine input, seawater upwelling and metabolic activity of plankton communities, environmental variability in coastal water are larger than in open-ocean ecosystems (Duarte and Cerbrian, 1996).' in **lines 313–315.**

Lns 91-92: Would the authors consider adding 'geographically-distinct' strains to this line to emphasize both the importance of their own insights and the more general need to consider different strains of other widespread species.
Response: Hence, multiple strains, ideally from geographically distinct regions should be considered for investigating phytoplankton responses to climate change (Zhang et al., 2014; Blanco-Ameijeiras et al., 2016; Krumhardt et al., 2017). These changes are in **lines 91–94.**

Krumhardt, K. M., Lovenduski, N. S., Debora Iglesias-Rodriguez, M., and Kleypas, J. A.: Coccolithophore growth and calcification in a changing ocean, Prog. Oceanogr., 159, 276–295.

Lns 103-104: A plastic response also allows a strain to acclimate across an environmental gradient and widen its bio-geographical distribution. Rather than focus on just environmental change, what about environmental variability.
Response: Plasticity can be assessed by analyzing the reaction norm of one trait and a plastic response may allow a strain to acclimate across an environmental gradient and widen its bio-geographical distribution (Reusch, 2014; Levis and Pfennig, 2016). These changes are in **lines 103–106.**

Ln 126: How were all strains characterized and confirmed to be morphotype A (i.e. Distal shield length? Central area characteristics?)?

Response: Morphotype A was confirmed by scanning electron microscope.

All 17 strains belong to morphotype A (determined by scanning electron microscopy) and have been deposited in the Roscoff culture collection (RCC). These changes are in **Lines 128–129**.

Ln 140-141: Is this statement ('the best compromise') appropriate based on the authors end conclusion that the low experiment temperature relative to optimum growth conditions for the Canary Islands strains led to their low growth (and POC production)? It seems to be a compromise that had a definitive influence on the end outcome of the experiments. Is it not simpler to just delete this section (from the point of 'which ..' to the end) and come back to this in the discussion?

Reponse: Monoclonal populations were always grown in sterile-filtered (0.2 μm diameter, Sartobran® P 300, Sartorius) artificial seawater medium (ASW) as dilute batch cultures at 200 μmol photons $m^{-2}$ $s^{-1}$ light intensity under a 16/8 h light/dark cycle (light period: 5:00 a.m to 9:00 p.m.) at 16 °C which we consider to be the best a compromise for the three different origins of the strains. These changes are in **lines 140–144.**

Our results showed that low incubation temperature led to low growth and POC production rates of the Canary Islands population. In the discussion section, we compared influence of temperature on physiological rate of three populations. In an earlier study (Zhang et al., 2014), growth rates of the same Azores and Bergen strains as used here were measured at 8–28 °C. While at 26–28 °C the Bergen strains grew slower than the Azores strains, at 8 °C the Azores strains grew slower than the Bergen strains. This illustrates nicely that local temperature adaptation can significantly affect growth of *E. huxleyi* strains in laboratory experiments. Considering these findings and the temperature ranges of the three isolation locations (Table S1), the incubation temperature of 16 °C used in the present study was lower than the minimum sea surface temperature (SST) commonly recorded at the Canary Islands. In contrast, SSTs of 16 °C and lower have been reported for Azores and Bergen waters (Table S1). When exposed to 16 °C, growth rate of the Canary Islands population might have been already below their optimum and hence significantly reduced in comparison to the other populations (Fig. 2d). These changes are in **lines 324–336.**

Lns 152-153 (cf Lns 174-175): How were initial cell densities measured/estimated?

Response: (**In line 156**) There was 590 ml seawater in the 500 ml glass bottles. Before cells were inoculated to new seawater, finial cell concentrations (C0) were measured, and we calculated the inoculated volumes (V) according to V = (200 cell/ml x 590 ml)/C0. By using this method, we think that the initial cell concentration was 200 cell/ml.

Initial cell concentration was 200 cells $ml^{-1}$ (estimated from measured pre-culture concentrations and known dilution) and final cell concentration was lower than 100,000 cells $ml^{-1}$. These changes are in lines **155–157.**

Lns 289-290: An important result that should be emphasized in the abstract and conclusions.

Response: In the abstract, we added this content **in lines 45–46:** Our results indicate adaptation of *E. huxleyi* to their local environmental conditions and the existence of distinct *E. huxleyi* populations.

In the conclusion: we added this sentence **in lines 420–423:** The existence of distinct *E. huxleyi* populations and phenotypic plasticity of individual strains may both be important for *E. huxleyi* when adapting to natural environmental variability and to ongoing climate changes.

Lns 322-324: Suggest deleting 'causes' from this sentence.

Response: **For:** we delete these contents '*One of the reasons may be that compared to the Azores and Bergen populations, 16 ºC likely causes lower the carbon uptake and carbon-use efficiency of the Canary Islands population (Sett et al., 2014).*' in **lines 345–348**

Ln 351-352: Another potentially important conclusion, especially given the emphasis on determining time-dependent (or space-dependent) variations in coccolith-specific PIC quotas. However, the current paper lacks any details of the strain-specific variability in PIC quota and to what extent the different trends in pCO2-sensitivity (e.g. Fig. 3e) are driven by changes in growth rate and/or cellular (or coccolith) specific PIC quota. Can strain-specific information on PIC quota be added to the supplementary material to support this point with experimental data?

Response: PIC quota of population is shown in figure S2, and PIC quota of individual strain is shown in Figure S4. We measured PIC quota of individual strains at 11 $p$CO$_2$ levels **with no replicate.** This is the reason that we did not discuss PIC quota of individual strains.

We deleted this sentence **in lines 391–393:** '*Additionally, our results also suggest that strain-specific PIC quota may be the basis of variation in coccoliths of E. huxleyi within the morphotype A (Fig. S3) (Young, 1994; Paasche, 2002).*'

Ln 374: A two line conclusion seems relatively short based on the significant statements made in the conclusions. Either expand or delete?

Response: We added main result**: '**The existence of distinct *E. huxleyi* populations and phenotypic plasticity of individual strains may both be important for *E. huxleyi* when adapting to natural environmental variability and to ongoing climate changes.' **in lines 420–423**

We added ', and CO$_2$ response was modulated by other environmental factors such as temperature and light intensity.' in **lines 425–426.**

**List of changes**

**Abstract**

Lines 26–27: add 'from different areas'

Line 27: delete 'population'

Lines 28–29: add '3 populations'

Line 29: delete '17 strains'

Line 30: add ': 6 strains', add ': 5 strains'

Line 31: add ': 6 strains'

Lines 32–33: change 'displayed the expected optimum curve responses to the $p$CO$_2$ gradient' to 'increased with rising $p$CO$_2$ levels, reached maximum and declined thereafter'

Line 36: change 'a' to 'the'

Line 37: change 'fjord' to 'coast'

Line 38: add 'environmental variability including large', and delete 'variability'

Line 39: add 'fluctuations'

Line 41: add 'that'

Lines 42–43: change 'One of the reasons may be that the' to 'This pattern could be driven by temperature-CO$_2$-interactions where the'

Line 44: change 'is' to 'was'

Line 46: add 'and the existence of distinct *E. huxleyi* populations'

Lines 48–49: delete 'carbonate chemistry'

Line 49: add 'to changes in carbonate chemistry'

Line 50: add 'and adapt'

**Introduction**

Line 76: add '; Krumhardt et al., 2017'

Line 91: change 'These indicate that' to 'Hence,'

Lines 91–92: add ', ideally from geographically distinct regions'

Line 93: add ';'

Line 94: add 'Krumhardt et al., 2017'

Line 105: change 'to environmental change' to 'across an environmental gradient and widen its bio-geographical distribution'

**Materials and methods**

Lines 128–129: add '(determined by scanning electron microscopy)'

Line 129: change 'at' to 'in'

Lines 139–140: add '(Fig. S1)'

Line 143: change 'the best' to 'a'

Lines 156–157: add '(estimated from measured pre-culture concentrations and known dilution)'

Lines 202–206: add 'In a broad $p$CO$_2$ range, physiological rates are expected to initially increase quickly until reaching an optimum and then decline towards further increasing CO$_2$ levels (e.g. Krug et al. 2011). Hence we used the following modified Michaelis-Menten equation (Bach et al. 2011) which was fitted to measured cellular growth, POC and PIC production rates'

Lines 206–207: delete 'The nonlinear regression model (4) was used to fit growth, POC and PIC production rates'

Line 207: add 'and', and delete 'ing'

Line 211: delete 'is', add ',' and delete 'which indicates'

Line 212: delete 'the effect of', and add 'depicts the slope of the decline after optimum $CO_2$ levels in response to'

Line 213: delete 'the'

Line 214: add '(equation 5)', delete 'for physiological rates according to equation (5)', add 'and', and change 'M' to 'm'

Line 215: delete 'were calculated by using equation (4) based on $K_m$.'

Line 216: add 'following Bach et al., (2011).'

**Discussion**

Lines 313–315: add 'In addition, due to riverine input, seawater upwelling and metabolic activity of plankton communities, environmental variability in coastal waters are larger than in open-ocean ecosystems (Duarte and Cerbrian, 1996).'

Line 317: change 'ed with' to 'ing'

Line 329: add 'the', and change 'ed' to 'ion'

Lines 334–336: change 'thus it grew slower than the other populations' to 'hence significantly reduced in comparison to the other populations'

Lines 337–345: add 'Furthermore, compared to the Canary Islands population, the Azores population had higher maximum growth and POC production rates, and similar optimum $CO_2$ for these physiological rates. Again, this might be related to sub-optimal incubation conditions as temperature has been found to significantly modulate $CO_2$ responses in coccolithophores in terms of maximum rates, $CO_2$ optima and half-saturation, and $H^+$ sensitivity (De Bodt et al., 2010; Sett et al., 2014; Gafar et al., 2018; Gafar and Schulz, 2018). In a similar fashion light can also modulate $CO_2$ responses, hence different requirements by strains adapted to different light availabilities could also explain our observations (Zhang et al., 2015; Gafar et al., 2018; Gafar and Schulz, 2018).'

Lines 345–348: delete 'One of the reasons may be that compared to the Azores and Bergen populations, 16 °C likely causes lower the carbon uptake and carbon-use efficiency of the Canary Islands population (Sett et al., 2014).'

Lines 350–355: add 'In addition, the Canary Islands population showed smallest variability in optimum $p CO_2$ and maximum values for growth and POC production rates (Fig. 2). The reason may be that low incubation temperature predominantly limited growth and POC production rates of the Canary Islands population, and decreased the sensitivities of these physiological rates to rising $p CO_2$.'

Line 365: delete 'reflected in'

Line 366: add 'supposed to be one reason for'

Lines 367–376: add 'The optimum temperature for growth of the Bergen population was about

℃ and was 5 ℃ higher than the maximum SST in Bergen waters (Zhang et al. 2014). Furthermore, in comparison to the Azores and Canary Islands populations, larger optimum $pCO_2$ of growth rate indicates that the Bergen population may benefit more from the rising $CO_2$ levels at increasing temperatures. PIC : POC ratios of the Azores and Bergen populations declined with rising $pCO_2$, whereas PIC : POC ratios of the Canary Islands population were rather constant (Fig. S6). As changes in PIC :POC ratios of coccolithophore blooms may impact on the biological carbon pump, different regions might see different changes in the future ocean.'

Line 377: change 'populations' to 'immigrant'

Line 378: add 'genotypes'

Lines 378–379: delete 'when having a higher potential to adapt to a changing environment'

Line 380: change 'take up' to 'is thought to utilize', and change 'to calcify and' to 'for calcification which'

Line 381: add 's'

Lines 382–385: delete 'Thus, due to population-specific growth and PIC production rates or quotas, changes in species composition, corresponding changes in PIC productions, may affect the ability of the ocean to take up $CO_2$.'

Line 389: add 'acclimate and', and delete 'their'

Lines 390–391: add 'and potentially to attenuate the short-term effects of changing environments on fitness-relevant traits'

Lines 391–393: delete 'Additionally, our results also suggest that strain-specific PIC quota may be the basis of variation in coccoliths of *E. huxleyi* within the morphotype A (Fig. S3) (Young, 1994; Paasche, 2002).'

Line 396: add ',', delete 'and', and add 'and'

Line 397: add 'potentially forms the basis for selection'

Line 404: add 'a', and delete 's'

Line 406: add 'er', and add 'or other'

Line 407: add 'competitive abilities', add 's', and delete 'strains in the oceans'

Line 408: change 'S' to 'Further, a s'

Line 409: change '4' to '5', change 'suggests' to 'indicates', and change 'ed' to 'ing'

Line 410: change 'can' to 'will', add 'or fix', and delete 'from the oceans or'

Line 411: delete 'fix carbon faster', and change 'this' to 'When extrapolated to the ocean, *E. huxleyi* blooms'

Line 412: change 'or the' to 'and its'

Line 413: delete 'of the oceans when large *E. huxleyi* blooms occur'

Line 414: change 'will' to 'has the potential to'

**Conclusions**

Lines 420–423: add 'The existence of distinct *E. huxleyi* populations and phenotypic plasticity of individual strains may both be important for *E. huxleyi* when adapting to natural environmental variability and to ongoing climate changes.'

Line 424: change 'or' to 'and'

Lines 425–426: add ', and $CO_2$ response was modulated by other environmental factors such as temperature and light intensity.'

[revised manuscript text omitted]

Figure 1

[Figure]

Figure 2

[Figure]

Figure 3

---

## Author Response (AR2)

**Responses to comments**

Dear referee,

We thank you for your supportive comments on our manuscript. Our detailed response in blue text to your comments is attached. Changes to the manuscript text are underlined.

General comments

Overall I think that the manuscript is very close to being ready; I only have a few comments. The new schematic diagram helps but it is not quite clear to me yet, I would add more details. The conclusions and global extrapolations could be worked a little bit more, they basically only cite the Paasche paper. The PIC: POC ratio results were a great addition!

Response: Agreed. We have changed the schematic diagram (Fig. S1), which showed the experimental setup clearly now.

'In addition, these results will improve our understanding on variation in physiological responses of different *E. huxleyi* populations to climate change, and variation in production of different areas in future oceans.' This sentence was added in **lines 359–362**.

PIC : POC ratios of the Azores and Bergen populations declined with rising $p$CO$_2$, whereas PIC : POC ratios of the Canary Islands population were rather constant (Figs. S6, S7). This sentense has shown **in lines 366–368**.

[Figure]

Fig. S1 A flow chat for the experimental processes.

Technical comments

Line 47: suggest deleting or changing this last phrase, not a very good conclusive phrase and this is related to understanding the implications of your findings and global impacts. Perhaps you can say accounting for this variability is important to understand how or whether Ehux might adapt to rising CO2 levels.

Response: Agreed.

'The existence of distinct responses to changes in carbonate chemistry between and within populations will likely benefit *E. huxleyi* to acclimate and adapt to rising $CO_2$ levels in the oceans.' was changed to 'Accounting for this variability is important to understand how or whether *E. huxleyi* might adapt to rising $CO_2$ levels.' **in lines 47–50**.

Line 137 and 147: it does sound like you try to hold your TA constant, I don't understand why you said you didn't in your response to my first review….

Response: Yes. In our study, total alkalinity (TA) was constant, which was shown **in line 137**. To response to your first review, we said 'our $CO_2$ manipulations are mimicking ongoing ocean acidification where $CO_2$/pH and DIC changes **at constant TA.**'

Line 237: perhaps I would add "as expected" and add citations

Response: Agreed. We added 'As expected' **in line 236**.

Line 258-265: why do you think that there are differences in some sensitivity constants but not in rates?

Response: As shown **in lines 209–210**, '*s*, the sensitivity constant, depicts the slope of the decline after optimum $CO_2$ levels in response to rising $H^+$', which means that sensitivity constant is relevant to rising $H^+$. However, growth, POC and PIC production rates are relevant to $CO_2$ and $H^+$ concentrations, and other environmental factors such as temperature and light intensity. So sensitivity constants and rates could show different results.

Line 323: suggest you rephrase to "this illustrates how adaptation to local temperature can …" I would delete "nicely", unnecessary.

Response: Agreed. 'nicely that local temperature adaptation' was replaced by 'how adaptation to local temperature' **in lines 323–324**.

Line 338: add comma after "fashion,"

Response: Agreed. Comma was added after 'fashion' **in line 338**.

Line 340-343: your conclusion is based on the assumption that temperature doesn't change but temperature will increase perhaps even faster than CO2 and this might have a greater impact. Plus, there could be interactions between temperature and CO2…

Response: Agreed. Compared to $CO_2$ concentrtion, temperature might have a greater impact on growth, POC and PIC production rates of *E. huxleyi*. However, our results cannot show this idea.

Temperature and $CO_2$ may interactively affect growth, POC and PIC production rates. In this study, incubation temperature (16 $^o$C) may predominantly limit physiological rates of Canary Islands populations. So we said 'Thus, with rising $CO_2$, growth, photosynthetic carbon fixation and calcification rates of the Canary Islands population cannot increase as much as in the Azores and Bergen populations.' **in lines 340–343**.

Line 363: delete "at increasing temperatures" or change to "and increasing temperatures"
Response: Agreed. We deleted 'at increasing temperatures' **in line 366**.

Line 367: please elaborate
Response: 'As changes in PIC : POC ratios of coccolithophore blooms may impact on the biological carbon pump, different regions might see different changes in the future ocean.' was replaced by 'As changes in PIC : POC ratios of coccolithophore blooms were suggested to impact on biological carbon pump (Rost and Riebesell, 2004), variation in PIC : POC ratios of different populations indicates that different regions might have different changes in marine carbon cycle in the future ocean'. These changes are in **lines 368–374**.

Line 370-72: consider more implications
Response: Agreed. We added 'and then give a positive feedback to rising atmosphare $CO_2$ levels' **in line 379**.

Line 392-95: I disagree, this is just a simple correlation, it doesn't say anything about dominance of strains. My take is that higher growth rate, means larger population and so, greater production.
Response: Agreed. We changed 'the dominating strains will also take up or fix dissolved inorganic carbon faster' to 'higher grwoth rate means larger populations and then greater production'. These changes are **in lines 402–404**.

Line 395-396: again, the conclusions need more work
Response: We deleted this sentense 'When extrapolated to the ocean, *E. huxleyi* blooms may increase the potential of the oceans to absorb $CO_2$ from the atmosphere and its carbon storage capacity (Blanco-Ameijeiras et al., 2016), which has the potential to mitigate rising $CO_2$ levels in the atmosphere.' **in lines 404–407**.

Line 408: I would say something like: in this case, we only studied the effects of rising CO2 but future studies should take into account simultaneous effects from other interacting factors such as light and temperature variability.
Response: Agreed. We changed these contents ', and $CO_2$ response was modulated by other environmental factors such as temperature and light intensity' to 'In this study, we only studied the effects of rising $CO_2$ but future studies should take into account simultaneous effects from other interacting factors such as light and temperature variability.'. These changes are **in lines 417–421**.

**List of changes**

Lines 47–50: changed 'The existence of distinct responses to changes in carbonate chemistry between and within populations will likely benefit *E. huxleyi* to acclimate and adapt to rising $CO_2$ levels in the oceans.' to 'Accounting for this variability is important to understand how or whether *E. huxleyi* might adapt to rising $CO_2$ levels.'

Line 236: added 'As expected' and changed 'G' to 'g'.

Lines 323–324: changed 'nicely that local temperature adaptation' to 'how adaptation to local temperature'.

Line 337: deleted 'De Bodt et al., 2010;'

Line 338: added ','.

Lines 359–362: added 'In addition, these results will improve our understanding on variation in physiological responses of different *E. huxleyi* populations to climate change, and variation in production of different areas in future oceans.'.

Lines 366: deleted 'at increasing temperatures'.

Line 368: added 's' and ', S7'.

Lines 368–374: changed 'As changes in PIC : POC ratios of coccolithophore blooms may impact on the biological carbon pump, different regions might see different changes in the future ocean.' to 'As changes in PIC : POC ratios of coccolithophore blooms were suggested to impact on biological carbon pump (Rost and Riebesell, 2004), variation in PIC : POC ratios of different populations indicates that different regions might have different changes in marine carbon cycle in the future ocean.'.

Line 379: added 'and then give a positive feedback to rising atmosphare $CO_2$ levels'.

Lines 402–404: changed 'the dominating strains will also take up or fix dissolved inorganic carbon faster' to 'higher grwoth rate means larger populations and then greater production'.

Lines 404–407: deleted 'When extrapolated to the ocean, *E. huxleyi* blooms may increase the potential of the oceans to absorb $CO_2$ from the atmosphere and its carbon storage capacity (Blanco-Ameijeiras et al., 2016), which has the potential to mitigate rising $CO_2$ levels in the atmosphere.'.

Lines 417–421: changed ', and $CO_2$ response was modulated by other environmental factors such as temperature and light intensity.' to 'In this study, we only studied the effects of rising $CO_2$

but future studies should take into account simultaneous effects from other interacting factors such as light and temperature variability.'.

Line 564: added ', doi: 10.1016/j.pocean.2017.10.007, 2017'.

[revised manuscript text omitted]

Figure 1

[Figure]

Figure 2

[Figure]

Figure 3